

# Assessing the sea ice microwave emissivity up to submillimeter waves from airborne and satellite observations

Nils Risse[1], Mario Mech[1], Catherine Prigent[2], Gunnar Spreen[3], and Susanne Crewell[1]

[1]Institute for Geophysics and Meteorology, University of Cologne, Cologne, Germany
[2]Laboratoire d'Etudes du Rayonnement et de la Matière en Astrophysique et Atmosphères, Observatoire de Paris, CNRS, Paris, France
[3]Institute of Environmental Physics, University of Bremen, Bremen, Germany

**Correspondence:** Nils Risse (n.risse@uni-koeln.de)

**Abstract.** Upcoming submillimeter wave satellite missions require an improved understanding of the sea ice emissivity to separate atmospheric and surface microwave signals under dry polar conditions. This work investigates hectometer-scale airborne sea ice emissivity observations between 89 and 340 GHz combined with high-resolution visual imagery from two Arctic airborne field campaigns in summer 2017 and spring 2019 northwest of Svalbard, Norway. We identify four distinct sea ice emissivity spectra through K-Means clustering, which occur predominantly over multiyear ice, first-year ice, young ice, and nilas. Nilas features the highest, and multiyear ice features the lowest emissivity among the clusters. Each cluster exhibits similar nadir emissivity distributions from 183 to 340 GHz. To relate hectometer-scale airborne to kilometer-scale satellite footprints, we quantify the reduction of airborne emissivity variability with increasing footprint size. At 340 GHz, the emissivity interquartile range decreases by almost half from the hectometer scale to a footprint of 16 km, typical for satellite instruments. Furthermore, we collocate the airborne observations with polar-orbiting satellite observations. After resampling, the absolute relative bias between airborne and satellite emissivities at similar channels lies below 3 %. Additionally, spectral nadir emissivity variations on the satellite scale are low, with slightly decreasing emissivity from 183 to 243 GHz, which occurs for all hectometer-scale clusters except for predominantly multiyear ice. Our results will enable the development of microwave retrievals and assimilation over sea ice from current and future satellite missions such as Ice Cloud Imager (ICI) and European Polar System (EPS) Sterna.

## 1 Introduction

Passive microwave observations from polar-orbiting satellites continuously monitored polar regions with high spatial coverage for over five decades (Comiso and Hall, 2014). They are essential for atmosphere (e.g., Triana-Gómez et al., 2020; Perro et al., 2020) and sea ice (e.g., Spreen et al., 2008; Kilic et al., 2020; Soriot et al., 2023) or joint atmosphere–sea ice retrievals (e.g., Scarlat et al., 2020; Rückert et al., 2023; Kang et al., 2023). Such satellite-based retrievals help to understand the accelerated



Arctic near-surface warming compared to the global mean (Rantanen et al., 2022; Wendisch et al., 2023). However, the highly variable sea ice emission causes uncertainties in satellite retrievals and severely limits the use of surface-sensitive microwave channels in operational numerical weather prediction data assimilation compared to open ocean (Lawrence et al., 2019). There-
fore, current research aims to improve the assimilation of microwave observations over sea ice, e.g., Bormann (2022) showed improved performance when assuming Lambertian instead of specular reflection in forward simulations.

Further spaceborne capabilities will become available through the novel Ice Cloud Imager (ICI; Buehler et al., 2007) and European Polar System (EPS) Sterna (Albers et al., 2023) instruments, with first-time operational channels above 200 GHz. These channels provide higher sensitivity to small cloud ice particles than the current passive microwave sensors (Buehler
et al., 2012; Wang et al., 2017a; Eriksson et al., 2020). However, variable emission of polar surfaces also adds significantly to the atmospheric signal received at the 243 (only ICI) and 325 GHz channels due to the dry atmosphere (Wang et al., 2017b).

While considerable interest exists in expanding the sea ice emissivity estimates to submillimeter waves, few field observations cover this frequency range. The first brightness temperature (TB) observations of sea ice at 220 GHz were obtained using an airborne cross-track scanning radiometer (Hollinger et al., 1984). However, the sea ice emissivity was derived only
at lower frequencies up to 140 GHz due to high TB noise and low atmospheric transmissivity at 220 GHz during the field study. The observations revealed similar nadir emissivities at 90 and 140 GHz with higher emissivity over young ice (0.96) and lower emissivity over multiyear ice (0.68). Airborne observations with along-track scanning radiometers by Hewison and English (1999) provide detailed emissivity spectra for typical sea ice types and snow from 24 to 157 GHz and demonstrate the importance of volume scattering within snow at 157 GHz. Hewison et al. (2002) calculated nadir emissivities up to 183 GHz
of sea ice with different development stages from new to multiyear ice with similar instrumentation as in Hewison and English (1999). Haggerty and Curry (2001) observed first-time emissivities up to 243 GHz at nadir at about 1 $\mathrm{km}^2$ resolution. However, leads, which are typically smaller, could not be resolved. The 340 GHz channel on board the same aircraft was insensitive to surface emission due to the low atmospheric transmissivity. Airborne observations in Wang et al. (2017b) measured sea ice emissivities up to 325 GHz that reveal high spatial variability, but the driving sea ice properties at this frequency could not be
estimated.

While the field studies demonstrate the high sensitivity of microwaves to sea ice and snow properties for limited regions, only global sea ice emission information allows for atmospheric retrievals from satellites. As modeling of sea ice emission is computationally expensive and requires detailed knowledge of the sea ice and snow properties (Royer et al., 2017; Picard et al., 2018; Rückert et al., 2023), which is missing on global scales, spaceborne emissivity climatologies were developed
(Wang et al., 2017b; Munchak et al., 2020). The Tool to Estimate Land Surface Emissivity from Microwave to Submillimeter Waves (TELSEM$^2$; Wang et al., 2017b) climatology for sea ice and land surfaces additionally extrapolates the emissivity up to 700 GHz to provide first-guess emissivities for upcoming satellite missions such as ICI. To simultaneously retrieve atmospheric and sea ice and snow properties, radiative transfer models of sea ice and atmosphere are combined (Rückert et al., 2023; Kang et al., 2023). Kang et al. (2023) additionally simulated sea ice growth to increase the temporal consistency of the retrieved
sea ice and snow properties. However, the sea ice radiative transfer models might only be valid below 100 GHz. Recently, observed snow emissivities up to 243 GHz were successfully simulated based on realistic snow properties (Wivell et al., 2023).



This result highlights the need for similar sea ice emissivity field observations up to submillimeter waves to improve future modeling studies of sea ice. These field observations must also be related to satellite observations, which resolve the surface at a much coarser resolution.

The limitation of sea ice emissivity observations at submillimeter waves and its relevance for future satellite missions motivates our study, which is structured around two objectives. First, we aim to identify critical physical sea ice and snow properties affecting emissivity up to submillimeter waves observed during two airborne field campaigns. We calculate the sea ice emissivity from TBs at 89 (25° incidence angle, horizontal polarization), 183, 243, and 340 GHz (nadir) from the airborne Microwave Radar/Radiometer for Arctic Clouds (MiRAC; Mech et al., 2019). Then, we characterize typical emissivity spectra with airborne visual imagery and surface temperature observations. Second, we aim to relate the observed hectometer-scale emissivity observations to the satellite scale. This includes an assessment of the emissivity variability as a function of footprint size. Furthermore, we collocate MiRAC with observations from polar-orbiting satellites and analyze the spectral emissivity variations observed at satellite resolution from 89 to 340 GHz relevant for upcoming satellite missions such as ICI and EPS Sterna.

The manuscript is outlined as follows. Section 2 describes the airborne field data, microwave instruments, and auxiliary data. Section 3 details the emissivity calculation. Section 4 identifies relevant sea ice and snow properties that affect emissivity from airborne observations. The comparison of emissivity between airborne and satellite observations is presented in Sect. 5 before the study is summarized and concluded in Sect. 6.

## 2 Data

### 2.1 Field data

We use airborne observations from two campaigns, i.e., Arctic Cloud Observations Using Airborne Measurements during Polar Day (ACLOUD) from 23 May to 26 June 2017 (Wendisch et al., 2019; Ehrlich et al., 2019b) and Airborne Measurements of Radiative and Turbulent Fluxes of Energy and Momentum in the Arctic Boundary Layer (AFLUX) from 19 March to 11 April 2019 (Mech et al., 2022a). Both campaigns were conducted as part of the Arctic Amplification: Climate Relevant Atmospheric and Surface Processes, and Feedback Mechanisms ((AC)[3]) research project (Wendisch et al., 2023). The research flights (RFs) with the *Polar 5* aircraft (Wesche et al., 2016) from the Alfred Wegener Institute Helmholtz Centre for Polar and Marine Research (AWI) covered the Fram Strait northwest of Svalbard, Norway (Fig. 1). Various sea ice characteristics were observed during ACLOUD, i.e., RF23 on 25 June and RF25 on 26 June 2017, and AFLUX, i.e., RF08 on 31 March, RF14 on 8 April, and RF15 on 11 April 2019, under clear-sky conditions over sea ice suitable for emissivity estimation. During the two ACLOUD flights, melt ponds formed on the sea ice with open water between individual ice floes. During the three AFLUX flights, snow-covered sea ice, mostly multiyear ice (Fig. A1), prevailed with nilas in refrozen leads between individual ice floes. Higher fractions of open water during AFLUX were observed only in the marginal sea ice zone of RF08. The infrared-based mean surface temperatures lie near the freezing point with 0.8 to 1°C during both ACLOUD flights and well below the freezing point with -22 to -17°C during the three AFLUX flights. The integrated water vapor is about 10 to 10.3 kg m$^{-2}$ during the two



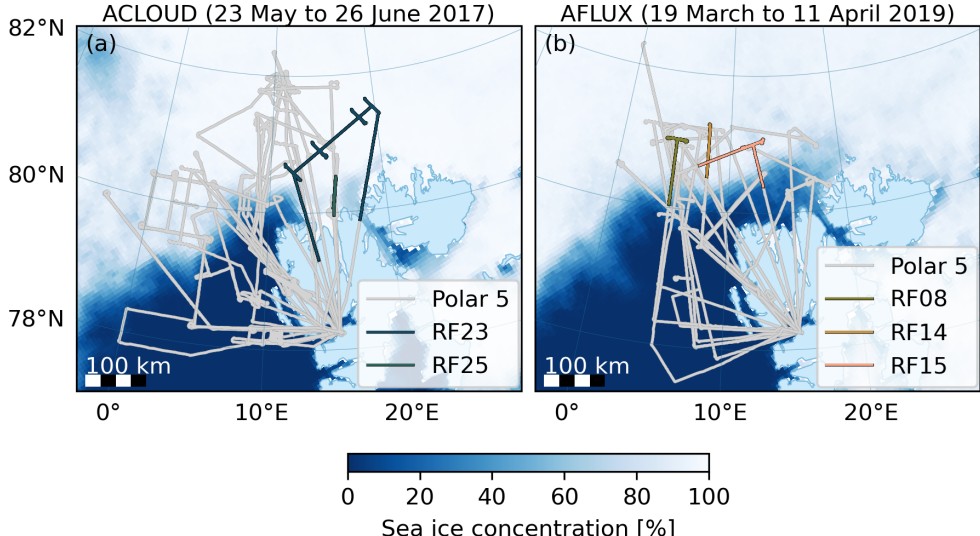

**Figure 1.** All *Polar 5* flights, clear-sky segments over sea ice, and mean sea ice concentration (Spreen et al., 2008) during (a) ACLOUD from 23 May to 26 June 2017 and (b) AFLUX from 19 March to 11 April 2019.

ACLOUD flights and 1.3 to 2 $\mathrm{kg\,m^{-2}}$ during the three AFLUX flights, which indicates reduced water vapor emissions and high atmospheric transmissivity during AFLUX.

## 2.2 Airborne microwave instruments

*Polar 5* carried the MiRAC package, which includes the combined active–passive component MiRAC-A mounted inside a belly pod underneath the aircraft's fuselage and the solely passive component MiRAC-P deployed inside the aircraft cabin (Mech et al., 2019). MiRAC-A consists of a 94 GHz cloud radar and a passive 89 GHz channel with horizontal polarization and measures with a 25° incidence angle backward. MiRAC-P measures at six double-side band water vapor channels (183.31±0.6, ±1.5, ±2.5, ±3.5, ±5.0, and ±7.5 GHz) and two window channels (243 and 340 GHz) at nadir (see Table 1). Both MiRAC components measure with a temporal resolution of 1 s. We omit MiRAC-A observations during low flights, i.e., with a slant path between the instrument and surface below 500 m, due to contamination from back-scattered broadband noise of the cloud radar. This threshold excludes MiRAC-A entirely during ACLOUD RF25 where the flight altitude during the clear-sky transects over sea ice ranges from 60 to 350 m. For the other four flights, the typical flight altitudes range from 60 m to 3 km with about 80 % (15 %) of the time below 500 m (above 2.5 km). Furthermore, we exclude observations with aircraft roll or pitch angles above 10°. The flight distance over which MiRAC provides emissivities depends on the channel, ranging from 400 km at 89 GHz to 1,700 km at 243 GHz. For about 200 km of this distance, all MiRAC-A and -P channels provide emissivities nearly instantaneously, i.e., the spatially matched footprint centers of MiRAC-P and the inclined MiRAC-A are less than 200 m apart.





**Table 1.** Specifications of MiRAC-A and -P channels.

| Instrument | Channel | Frequency [GHz] | Polarization | Incidence angle [°] | Field of view [°] |
|---|---|---|---|---|---|
| MiRAC-A | 1 | 89 | H | 25 | 0.85 |
| MiRAC-P | 1 | 183.31±0.6 | V | 0 | 1.3 |
| | 2 | 183.31±1.5 | V | 0 | 1.3 |
| | 3 | 183.31±2.5 | V | 0 | 1.3 |
| | 4 | 183.31±3.5 | V | 0 | 1.3 |
| | 5 | 183.31±5.0 | V | 0 | 1.3 |
| | 6 | 183.31±7.5 | V | 0 | 1.3 |
| | 7 | 243 | H | 0 | 1.25 |
| | 8 | 340 | H | 0 | 1.0 |

The instrument receivers were calibrated with a two-point calibration using liquid nitrogen and an internal target at the beginning of each campaign. In addition, MiRAC-A performed gain calibrations every 15 min, and MiRAC-P every 20 min during flights using an internal target. After the campaign, we applied a bias correction of the 89 GHz TBs based on Passive and Active Microwave Radiative Transfer (PAMTRA; Mech et al., 2020) forward simulations by using dropsonde profiles under clear-sky conditions over the open ocean extended by ERA5 reanalysis (Hersbach et al., 2020) to the top of the atmosphere and a sea surface temperature analysis (UK Met Office, 2012) as input. The added 89 GHz TB offset for the ACLOUD (AFLUX) flights in this study is 11 (32) K and decreases linearly towards higher TBs. This high calibration offset occurred due to difficult weather conditions during the liquid nitrogen calibration. We estimate the accuracy of the offset correction to be 2 K. For MiRAC-P, no such calibration issues occurred due to its location inside the aircraft cabin (Mech et al., 2019). The TB noise is about 0.5 K for MiRAC-A (Küchler et al., 2017) and -P (Mech et al., 2019). Hence, we assume the overall TB uncertainty from bias correction and noise to be 2.5 K at 89 GHz and 0.5 K at all other frequencies. The footprint size at 60 ms$^{-1}$ flight velocity with 1 s integration time is about 70×130 m$^2$ at 3 km and 1×60 m$^2$ at 60 m flight altitude at 183 GHz, i.e., at nadir with an opening angle of 1.3° (see Table 1). We shift MiRAC's measurement time by 1 to 2 s (2 to 5 s) during ACLOUD (AFLUX) relative to the infrared radiometer KT-19 based on lagged correlations between 243 GHz TBs and KT-19 infrared TBs during the clear-sky sea ice emissivity flight segments. Note that the 243 GHz channel showed the highest correlation of all MiRAC-P channels with the infrared TB during both campaigns due to its high atmospheric transmission compared to the other MiRAC-P channels.



## 2.3 Satellite microwave instruments

We focus on commonly used cross-track and conical scanning polar-orbiting microwave radiometers. These are the Microwave
Humidity Sounder (MHS; EUMETSAT, 2010), the Advanced Technology Microwave Sounder (ATMS; Kim et al., 2014), the
Special Sensor Microwave Imager/Sounder (SSMIS; Kunkee et al., 2008), and the Advanced Microwave Scanning Radiometer
2 (AMSR2; JAXA, 2016) with their platforms and specifications summarized in Table 2. To ensure consistency among the
sensors, we use inter-calibrated Level 1C TB data (NASA Goddard Space Flight Center and GPM Intercalibration Working
Group, 2022). This inter-calibration corrects offsets between the constellation satellites using the well-calibrated Global Pre-
cipitation Mission (GPM; Hou et al., 2014) Microwave Imager (GMI), which covers up to 65° N, as a reference (Berg et al.,
2016).

MHS and ATMS scan cross-track at incidence angles up to 59 and 64°, respectively, and SSMIS and AMSR2 scan conically
at an incidence angle of 53 and 55°, respectively. MHS and ATMS measure TBs with nominal vertical (QV) or horizontal
polarization (QH) at nadir rotating with view angle $\alpha$ as

$$T_{b,\text{QV}} = T_{b,\text{V}} \cos^2(\alpha) + T_{b,\text{H}} \sin^2(\alpha) \tag{1}$$

and

$$T_{b,\text{QH}} = T_{b,\text{H}} \cos^2(\alpha) + T_{b,\text{V}} \sin^2(\alpha). \tag{2}$$

We use MHS and ATMS only with incidence angles from 0 to 30° because these provide similar observing conditions as
MiRAC. Moreover, fewer footprints with higher incidence angles collocate with MiRAC, and their increased footprint sizes
make comparisons more uncertain. With this incidence angle filter for MHS and ATMS, the footprint sizes are mostly around
$16 \times 16$ km$^2$ with the highest resolution of $3 \times 5$ km$^2$ by AMSR2. MHS on board the NOAA-18 spacecraft operated only
during ACLOUD, and the Metop-C and NOAA-20 spacecraft operated only during AFLUX. The 150 GHz channel of SSMIS
on board the DMSP-F18 was not available due to its failure (Berg et al., 2016).

MiRAC overlaps spectrally with MHS, ATMS, and SSMIS at 89 and 183 GHz and AMSR2 at 89 GHz. However, MiRAC's
89 GHz channel with horizontal polarization at 25° is not directly comparable with the satellite channels because MHS and
ATMS measure mostly vertically polarized TB near this incidence angle, and SSMIS and AMSR2 measure at higher incidence
angles. Only MiRAC's 183 GHz near-nadir channel is directly comparable with near-nadir observations from MHS and ATMS.

## 2.4 Ancillary observations

The emissivity retrieval requires ancillary information on the atmospheric thermodynamic profile and surface temperature. We
construct the thermodynamic profile below 3 km altitude from measurements of the aircraft's nose boom and dropsondes and
above 3 km altitude from radiosondes (Maturilli, 2020) launched at the AWIPEV station operated jointly by AWI and the
Polar Institute Paul Emile Victor (IPEV) in Ny-Ålesund, Svalbard, Norway (Neuber, 2003). We assume constant temperature
and humidity from the lowest flight altitude down to the surface if no dropsonde information is available over sea ice. The



**Table 2.** Specifications of MHS, ATMS, SSMIS, and AMSR2 channels used in this study. The instantaneous field of view (IFOV) of MHS and ATMS is given for nadir. The polarizations of MHS and ATMS are either nominal vertical at nadir rotating with view angle (QV) or nominal horizontal at nadir rotating with view angle (QH). The polarizations of SSMIS and AMSR2 are either horizontal (H) or vertical (V). Only 0 to 30° incidence angles from MHS and ATMS are used here.

| Instrument | Channel | Frequency [GHz] | Polarization | Incidence angle [°] | IFOV [km$^2$] |
|---|---|---|---|---|---|
| MHS (Metop-A, -B, -C$^a$, NOAA-18$^b$, -19) | 1 | 89 | QV | 0–30 | 16×16 |
| | 2 | 157 | QV | 0–30 | 16×16 |
| | 5 | 190.31 | QV | 0–30 | 16×16 |
| ATMS (SNPP, NOAA-20$^a$) | 16 | 88.2 | QV | 0–30 | 32×32 |
| | 17 | 165.5 | QH | 0–30 | 16×16 |
| | 18 | 183.31±7 | QH | 0–30 | 16×16 |
| SSMIS (DMSP-F16, -F17, -F18) | 17 | 91.655 | V | 53 | 9×15 |
| | 18 | 91.655 | H | 53 | 9×15 |
| | 8 | 150 | H | 53 | 9×15 |
| | 9 | 183.31±6.6 | H | 53 | 9×15 |
| AMSR2 (GCOM-W1) | 13 | 89 | V | 55 | 3×5 |
| | 14 | 89 | H | 55 | 3×5 |

$^a$ Satellite operated only during AFLUX

$^b$ MHS on board NOAA-18 operated only during ACLOUD

uncertainties of temperature and relative humidity are ±0.2 K and ±2 % for dropsondes (Vaisala, 2010), ±0.2–0.4 K and

±3–4 % for radiosondes (Maturilli, 2020), and ±0.3 K and ±0.4 % for the nose boom (Ehrlich et al., 2019b).

The airborne KT-19 provides infrared TBs integrated over the atmospheric window from 9.6 to 11.5 μm with 1 s resolution under an opening angle of 2°. Hence, its opening angle is slightly higher than MiRAC's opening angles. The accuracy of KT-19 is about ±0.5 K. The infrared TB is converted to surface skin temperatures with an infrared emissivity of 0.995, which approximates the infrared emissivity of typical sea ice types with an accuracy of 0.01 to 0.02 (Hori et al., 2006). We also

require an accurate description of the surface temperature at the satellite footprint scale. Therefore, we use the daily Level 4 Arctic sea and ice surface temperature reanalysis with a resolution of 0.05×0.05° (Nielsen-Englyst et al., 2023), hereafter referred to as NE23. A comparison between the airborne surface temperature based on KT-19 and the NE23 temperature reveals biases of 4 to 6 K during ACLOUD and -1 to 1 K during AFLUX (KT-19 minus NE23). During ACLOUD, the KT-19 temperatures lie close to the melting point, which agrees with observed melting conditions with a snow liquid water fraction

around 15 % (Rosenburg et al., 2023). We use the nearest NE23 ice surface temperature pixel to the satellite footprint as input to the sea ice emissivity calculation. Furthermore, a downward-looking camera equipped with a fish-eye lens operating in the



visible spectrum (red, green, and blue) on board *Polar 5* provides information on the sea ice characteristics every 4 to 6 s. Finally, three data products add surface information, i.e., daily sea ice concentration maps of the University of Bremen with $6.25{\times}6.25\ \mathrm{km}^2$ resolution based on AMSR2 (Spreen et al., 2008), daily multiyear ice concentration maps of the University of

Bremen with $12.5{\times}12.5\ \mathrm{km}^2$ resolution based on AMSR2 and the Advanced Scatterometer (ASCAT; Melsheimer and Spreen, 2022), and Sentinel-2B Level 2A visual images with $20{\times}20\ \mathrm{m}^2$ resolution (European Space Agency, 2021).

We utilize topographic data from the Norwegian Polar Institute to exclude observations over land and near the coastline (Norwegian Polar Institute, 2014). Specifically, we exclude data within 150 m from the shoreline for MiRAC and within about one footprint radius of 2.5 km (8 km) for AMSR2 (MHS, ATMS, and SSMIS).

**2.5 Collocation of MiRAC with satellites**

To compare MiRAC with satellites, we need nearly simultaneous and spatially aligned observations. We ensure simultaneous observations by filtering collocations within a $\pm2$ h window, which maximizes the number of satellite overpasses and minimizes the effects of sea ice drift. Furthermore, we spatially align MiRAC with the nearest satellite footprints for each satellite overpass by imposing specific criteria: a footprint center distance threshold of about one footprint radius of 2.5 km (8 km)

for AMSR2 (MHS, ATMS, and SSMIS) and a minimum of 17 (50) MiRAC footprints within the AMSR2 (MHS, ATMS, and SSMIS) footprint. The latter criterion translates to a straight flight distance exceeding approximately 20 % of the satellite footprint diameter (10 % for ATMS at 89 GHz).

The number of satellite overflights with collocated footprints from MHS, ATMS, SSMIS, and AMSR2 is 15 (23), 0 (8), 11 (26), and 2 (9) during ACLOUD (AFLUX), respectively. We matched channels near 89 GHz with MiRAC-A and above

100 GHz with MiRAC-P. The number of satellite footprints collocated with MiRAC at 89 GHz during ACLOUD (AFLUX) is 87 (86), 0 (34), 108 (175), and 23 (159) for MHS, ATMS, SSMIS, and AMSR2, respectively. The number of satellite footprints collocated with MiRAC above 100 GHz during ACLOUD (AFLUX) is 222 (138), 0 (46), and 277 (261) for MHS, ATMS, and SSMIS, respectively. Around 70 MiRAC footprints occur within each of the satellite footprints at 89 GHz and about 200 above 100 GHz. The difference relates mainly to the higher resolution at 89 GHz for AMSR2.

**3 Methodology**

**3.1 Effective sea ice emissivity calculation**

We directly derive effective sea ice emissivity from observed clear-sky TBs and infrared-based skin temperature following Prigent et al. (1997). Typically, the skin temperature differs from the emitting sea ice or snow layer temperature (Tonboe, 2010). The depth of the emitting layer or penetration depth depends on sea ice and snow properties and decreases with increasing

frequency (Tonboe et al., 2006). Emissivities based on skin temperature are commonly referred to as effective emissivity, but hereafter, we use the term "emissivity" for better readability.





Harlow (2011) compared methods for estimating emitting layer temperature from 183 GHz observations. However, their applicability to our data is limited by the absence of simultaneous downwelling 183 GHz TB measurements and uncertainties in the atmospheric profile impacting surface temperature estimates. Other studies employ pre-calculated penetration depths and observed sea ice temperature profiles for specific ice types (Mathew et al., 2008, 2009), which does not apply to the diverse sea ice conditions presented here.

The emissivity calculation is based on the non-scattering radiative transfer (RT), valid under clear-sky conditions. The TB observed at aircraft or satellite height, denoted as $T_b$, is given by

$$T_b = T_s \cdot e \cdot t + T_b^{\downarrow} \cdot t \cdot (1 - e) + T_b^{\uparrow}, \tag{3}$$

with the surface emissivity $e$, surface temperature $T_s$, atmospheric transmissivity in viewing direction $t$, downwelling atmospheric radiation at the surface $T_b^{\downarrow}$, and upwelling atmospheric radiation at the observation height $T_b^{\uparrow}$. Solving Eq. (3) for the surface emissivity leads to

$$e = \frac{T_b - T_b^{\uparrow} - T_b^{\downarrow} \cdot t}{(T_s - T_b^{\downarrow}) \cdot t}. \tag{4}$$

Equation (4) can be solved using two RT simulations with $e = 0$ and $e = 1$ (Mathew et al., 2008)

$$e = \frac{T_b - T_b(e=0)}{T_b(e=1) - T_b(e=0)}. \tag{5}$$

We perform RT simulations for the *Polar 5* or satellite height with PAMTRA. In PAMTRA, we select the Rosenkranz (1998) gas absorption with modifications of the water vapor continuum absorption (Turner et al., 2009). We simulate specular and Lambertian reflections separately.

Satellite-based emissivity studies typically limit the emissivity calculation to channels with high atmospheric transmissivity. From aircraft, we can increase the transmissivity by flying at low altitudes. However, in addition to the transmissivity, the contrast between surface temperature and atmospheric downwelling TB dominates the surface sensitivity, i.e., the sensitivity of the observed TB to emissivity changes. This can be seen when calculating the partial derivative from Eq. (3) as

$$\frac{\partial T_b}{\partial e} = (T_s - T_b^{\downarrow}) \cdot t = T_b(e=1) - T_b(e=0). \tag{6}$$

This term equals the denominator in Eq. (5) and should be maximized to avoid noisy emissivity estimates. We identify 40 K as a reasonable threshold below which emissivity noise exceeds typical signatures of sea ice. Observed mean surface sensitivities during AFLUX are 200 K (50 K) at 89 GHz (340 GHz). Only 183 and 340 GHz observations during ACLOUD lie below the surface sensitivity threshold and are therefore excluded.

## 3.2 Emissivity uncertainty estimation

We estimate the emissivity uncertainty by propagating errors from TB (see Section 2.2), air temperature, relative humidity, and surface temperature. The assumed uncertainties for air temperature and relative humidity are $\pm 2$ K and $\pm 5$ %, respectively. This



assumed uncertainty is higher than the dropsonde and radiosonde uncertainty to account for the representability error along the flight path. The assumed uncertainty in surface temperature is ±3 K during ACLOUD and ±8 K during AFLUX. The surface temperature uncertainty for ACLOUD mainly accounts for errors in the infrared emissivity and KT-19 measurement uncertainty. The higher surface temperature uncertainty for AFLUX compared to ACLOUD accounts for the spread between
surface skin temperature and emitting layer temperature, which can deviate by up to 10 K over multiyear ice at 89 GHz due to insulating snow (Tonboe, 2010). During ACLOUD, we expect mostly isothermal sea ice due to surface melt (Perovich et al., 1997). The uncertainty estimation is performed only on aircraft and not on satellite observations.

## 3.3 Surface reflection model

The surface reflection model affects the direction from which downwelling atmospheric radiation is reflected at the surface.
Typically, the surface is approximated either purely specular or Lambertian. Over specular surfaces, the incidence angle matches the reflection angle, whereas Lambertian surfaces follow isotropic and unpolarized reflection. High sensitivity to the assumed surface reflection type occurs at nadir where MiRAC measures and low sensitivity at incidence angles between 50 to 60° where imagers like SSMIS and AMSR2 measure (Matzler, 2005; Karbou and Prigent, 2005).

Guedj et al. (2010) present a method to constrain the surface reflection model at 50 GHz sounding channels by combining TB
measurements with an emissivity retrieval. They calculate the emissivity at a wing channel of the absorption line to simulate an adjacent inner channel, which indicates Lambertian reflection over sea ice in winter and seasonal variations of specular contributions. Here, we adapt the method to 183 GHz MiRAC observations during the three AFLUX flights by following three steps. First, we calculate emissivities at 183.31±7.5 GHz under specular and Lambertian reflection. Second, we use the emissivities derived at 183.31±7.5 GHz to simulate TBs at 183.31±5 GHz with PAMTRA under the respective surface
reflection. Third, we compare the simulation with the observed TB at 183.31±5 GHz. The bias distribution lies closest to zero under the Lambertian assumption (Fig. 2). Despite the relatively high uncertainty near the water vapor absorption line, the results confirm Lambertian behavior of sea ice at 183 GHz as found by Harlow (2011) and Bormann (2022).

In the following, we present only emissivities calculated under Lambertian surface reflection from aircraft and satellites based on the findings at 183 GHz. Hence, we assume a similar surface reflection behavior at 89, 243, and 340 GHz. Addi-
tionally, we assume that the reflection type identified during AFLUX also represents ACLOUD observations where we lack 183 GHz surface emissivity. However, at 89 GHz, it is well known that sea ice exhibits a distinct polarization signature (NORSEX Group, 1983), indicating a specular contribution to the reflection. While we are still able to reproduce polarization signatures from satellites close to 50° incidence angle (Matzler, 2005), the specular contribution modifies the magnitude of simulated 25° reflected downwelling atmospheric TB. For MiRAC observations, fully specular emissivities are about 2 %
(0.6 %) higher than Lambertian emissivities of 0.7 (0.9) at 89 GHz under 25° incidence angle. This emissivity uncertainty is in the order of or lower than the uncertainty due to the surface temperature assumption at 89 GHz.





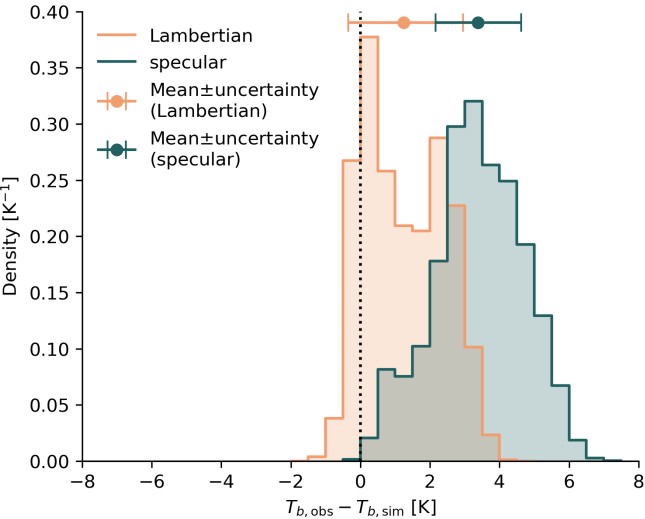

**Figure 2.** Histogram and mean of the difference between observed ($T_{b,\mathrm{obs}}$) and simulated 183.31±5 GHz TB ($T_{b,\mathrm{sim}}$) using 183.31±7.5 GHz emissivities under Lambertian and specular surface reflection during AFLUX. The TB bin width is 0.5 K.

## 4 Airborne observations

### 4.1 Case study

In this section, we first illustrate the available airborne observations along an 11 km transect during AFLUX RF08 on 31
March 2019 (Fig. 3). Satellite observations indicate about 75 % multiyear ice within the area (Fig. A1). The sea ice types along the transect are distinguishable in airborne visual camera observations (Fig. 3a–d) and Sentinel-2B imagery (Fig. 3e). We observe predominantly snow-covered sea ice over the transect's initial 7 km. Notably, surface structural variations from 3 to 4 km suggest the presence of young ice, which defines ice in the transition stage between nilas and first-year ice (Word Meteorological Organization, 2014), possibly formed within leads among the thicker multiyear ice. Progressing from 7 to
11 km, we encounter refrozen leads with nilas attached to individual snow-covered ice floes. The observed surface temperatures reflect the changing sea ice and snow properties with almost constant temperatures of -24°C over snow-covered sea ice and up to -18°C over nilas. The TBs vary much more than the 6 K in surface temperature with a range of 76, 47, 48, and 30 K at 89, 183.31±7.5, 243, and 340 GHz, respectively. This high variability demonstrates the importance of surface emissivity variations on the observed TB.

The difference between the minimum and maximum sea ice emissivity decreases with frequency, i.e., 0.35, 0.27, 0.24, and 0.21 at 89, 183.31±7.5, 243, and 340 GHz, respectively. The higher 89 GHz emissivity variability compared to the other frequencies likely relates to its horizontal polarization at 25° incidence angle. Previous studies show that horizontal polarization exhibits a higher sea ice emissivity variability at 89 GHz than vertical polarization at an incidence angle of 53° (e.g., Shokr et al., 2009). This relates to the enhanced sensitivity to sea ice and snow properties at horizontal polarization.



Similar effects likely occur at 25° incidence angle. Furthermore, the horizontal polarization at 89 GHz accounts for up to 0.05 lower emissivities compared to nadir depending on the sea ice type as shown by past airborne observations under varying incidence angles (Hewison and English, 1999). This partly explains the low 89 GHz emissivity observed here compared to the other nadir-viewing channels.

Despite the implications of incidence angle and polarization differences on spectral features, this transect showcases typical
sea ice emission signatures. Over nilas from 7 to 11 km, the sea ice emissivity increases at all channels with values between 0.9 and 1. Hewison and English (1999) and Hewison et al. (2002) observed similar emissivities at 89 and 183 GHz over bare ice under the same observing geometry as MiRAC. The sea ice emissivity over multiyear ice within the first 7 km is lower than over nilas at all frequencies. The snow-covered and refrozen lead from 3 to 4 km causes higher emissivities only at 89 GHz, likely due to the higher sensitivity to sea ice and snow properties at the horizontal polarization of this channel. Multiyear ice
observed at nadir in Hewison et al. (2002) lies close to multiyear ice observations along this transect. The 243 GHz nadir emissivity lies close to the mean emissivity of 0.84 at 220 GHz observed by Haggerty and Curry (2001). The consistency of MiRAC emissivity features with past sea ice emissivity studies provides confidence in the 243 GHz emissivity resolved at the hectometer scale. Moreover, the high similarity between 243 and 340 GHz emissivities shows that MiRAC provides submillimeter sea ice emissivities with a clear dependence on distinct sea ice types for the first time.

The ±8 K surface temperature uncertainty causes the highest emissivity uncertainty for all channels. The uncertainty magnitude varies highly between the channels. The lowest uncertainty range occurs at 89 and 243 GHz, and the highest at 183.31±2.5 GHz, which is the channel closest to the 183.31 GHz water vapor absorption line that exceeds the 40 K surface sensitivity threshold. In the following, we only show the 183.31±7.5 GHz channel due to its higher surface sensitivity and similar emissivity as the inner 183 GHz channels (Fig. 3j). The measured emissivity difference between multiyear ice and
nilas exceeds the emissivity uncertainty at all frequencies, while no significant variations occur in the first 7 km at 340 GHz. Overall, this case study demonstrates the relevance of sensitivity tests in interpreting the retrieved emissivities to distinguish emissivity features from uncertainties inherent to the assumptions of the emissivity calculation due to unknown sub-surface temperature and uncertain atmospheric thermodynamic profiles.

## 4.2 TB and emissivity variability

In the following, we analyze TB and emissivity distributions observed during all clear-sky flights over sea ice during ACLOUD and AFLUX (Fig. 4). The 89 GHz and 183 to 340 GHz histograms include different samples due to the exclusion of low flight altitudes at 89 GHz, which introduces a potential inconsistency (Table 3). Therefore, we compared these histograms with those from instantaneous measurements where all channels sample the same sea ice, and we found no significant changes in the shapes of the histograms (not shown). Hence, we present all the available observations here. The TB variability during
AFLUX exceeds that during ACLOUD at all frequencies (Fig. 4a–d). The 183, 243, and 340 GHz ACLOUD TBs show low variability and higher values due to the increased atmospheric water vapor and surface temperature. Two distinct peaks occur at 89 and 243 GHz during AFLUX. These peaks become even more pronounced in the emissivity distributions around 0.7 to 0.85 and 0.9 to 1, respectively (Fig. 4e–h). Their emissivity corresponds to multiyear ice and nilas in the AFLUX RF08





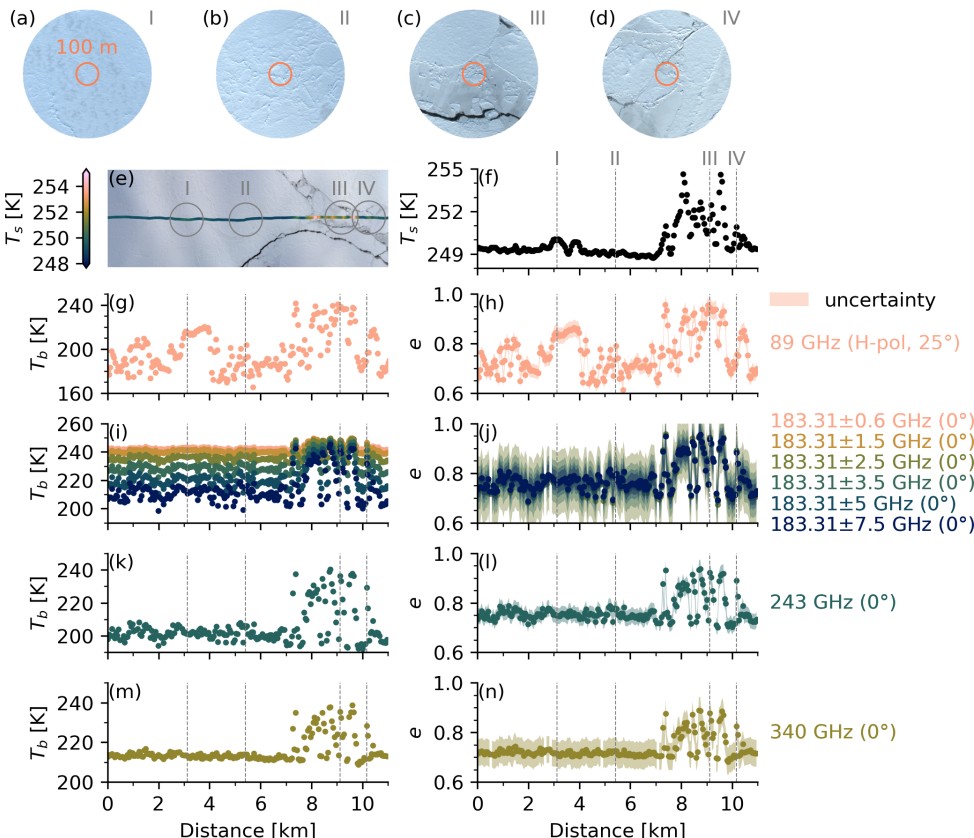

**Figure 3.** Observations along an 11 km transect from 4.28° E to 4.91° E at 81.01° N about 100 km north of the sea ice edge during AFLUX RF08 (31 March 2019). *Polar 5* flew westward (right to left here) at an altitude of 540 m within about 4 min starting at 11:39 UTC. (a–d) Fish-eye lens images with a 100 m diameter nadir reference circle at (I) 11:42:16 UTC, (II) 11:41:32 UTC, (III) 11:40:20 UTC, and (IV) 11:40:00 UTC. (e) Sentinel-2B L2A natural-color image at 14:37 UTC, flight track with surface skin temperature from KT-19, and location of the airborne imagery. (f) Surface skin temperature from KT-19. (g, i, k, m) TB at all MiRAC channels. (h, j, l, n) Emissivity and uncertainty from MiRAC's surface-sensitive channels, i.e., except for the two inner 183 GHz channels. The Sentinel-2B image was shifted by 2.5 km northward to correct for sea ice drift.

case (see Sect. 4.1). The histograms derived for 243 GHz are broader than the 220 GHz emissivities by Haggerty and Curry
(2001) due to MiRAC's higher resolution that captures previously unresolved leads. MiRAC's 340 GHz emissivity distribution follows a similar shape as the 183 and 243 GHz channels. The broader emissivity distribution at 340 GHz could be related to the higher emissivity uncertainty of 9 % compared to 183 GHz (6 %) and 243 GHz (5 %) during AFLUX (Table 3). The apparent shape difference of the 89 GHz distribution and the two times higher interquartile range (Table 3) indicate that this horizontally polarized and 25° inclined channel is more sensitive to sea ice and snow properties than the other channels at
higher frequencies. The narrower 89 GHz distribution during ACLOUD compared to AFLUX in the presence of melting sea



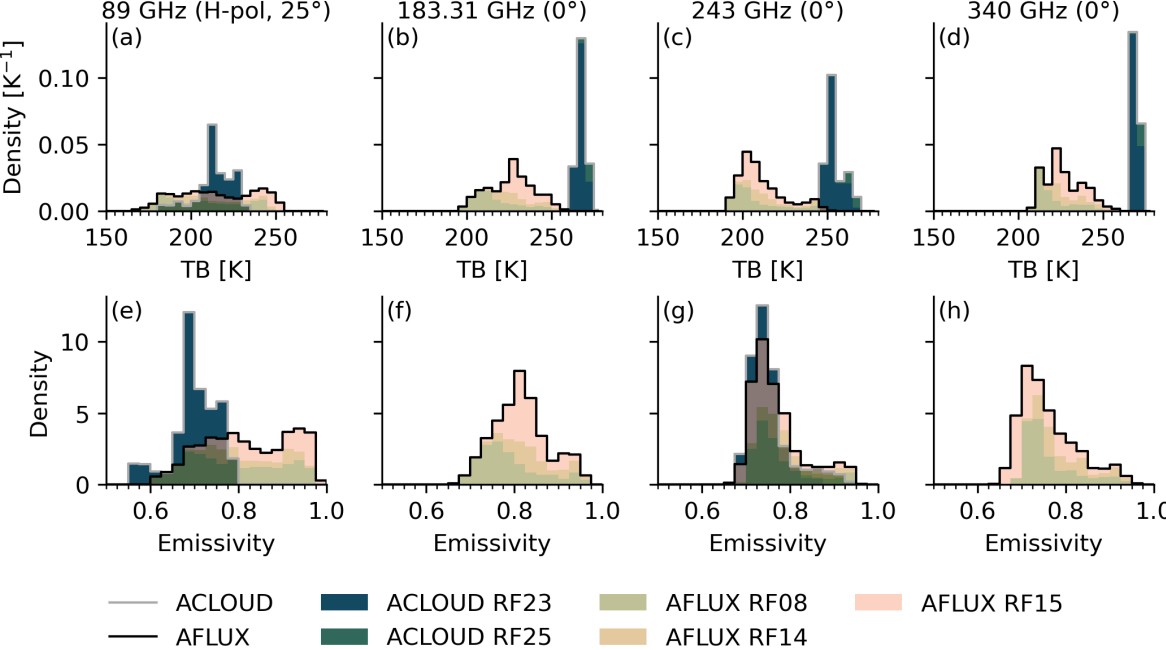

**Figure 4.** Histograms of (a–d) TB and (e–h) emissivity at (a, e) 89, (b, f) 183, (c, g), 243, and (d, h) 340 GHz during ACLOUD and AFLUX. The TB bin width is 5 K, and the emissivity bin width is 0.025. The 183 and 340 GHz observations during ACLOUD lie below the surface sensitivity threshold and are therefore excluded in (f) and (h). The histograms at 183, 243, and 340 GHz contain more samples than the 89 GHz histogram (see Table 3).

ice agrees with findings by Haggerty and Curry (2001). Lower emissivities during ACLOUD, i.e., the two lower peaks around 0.65, correspond to regions with lower sea ice concentrations. These emissivities should be treated with care due to the specular contributions of the sea surface.

### 4.3 Influence of sea ice and snow properties

In this section, we aim to relate the observed sea ice emissivity variability to sea ice and snow properties visible from fish-eye lens images and surface skin temperature. Previous airborne studies classified the sea ice based on airborne imagery or visual inspection and calculated emissivity spectra for each sea ice type (e.g., Hewison and English, 1999). However, this approach requires sea ice classification on a high temporal resolution. Therefore, we use K-Means clustering to extract distinct emissivity spectra similar to previous sea ice and snow emissivity studies (Wang et al., 2017b; Wivell et al., 2023). First, we perform a

normalization, subtracting the mean emissivity and dividing it by the standard deviation at each channel to ensure equal channel weighting. Then, we cluster the normalized emissivity spectra across all four MiRAC frequencies with K-Means to identify distinct sea ice emissivity spectra.



**Table 3.** Sea ice emissivity at MiRAC frequencies during individual flights and all flights of ACLOUD and AFLUX. Values denote the sample count (Cnt.), median (Mdn.), interquartile range (IQR), and mean relative uncertainty (Unc.). The 183, 243, and 340 GHz sample count is the same except for ACLOUD, where 183 and 340 GHz emissivities are not available.

| Campaign/RF | 89 GHz | | | | 183 GHz | | | | 243 GHz | | | 340 GHz | | |
|---|---|---|---|---|---|---|---|---|---|---|---|---|---|---|
| | Cnt. | Mdn. | IQR | Unc. [%] | Cnt. | Mdn. | IQR | Unc. [%] | Mdn. | IQR | Unc. [%] | Mdn. | IQR | Unc. [%] |
| ACLOUD RF23 | 3,431 | 0.7 | 0.06 | 2 | 15,152 | | | | 0.74 | 0.04 | 8 | | | |
| ACLOUD RF25 | | | | | 1,595 | | | | 0.87 | 0.06 | 5 | | | |
| AFLUX RF08 | 1,955 | 0.78 | 0.17 | 4 | 4,632 | 0.77 | 0.08 | 5 | 0.75 | 0.07 | 4 | 0.74 | 0.07 | 7 |
| AFLUX RF14 | 638 | 0.81 | 0.15 | 4 | 2,358 | 0.83 | 0.07 | 6 | 0.79 | 0.06 | 4 | 0.78 | 0.08 | 8 |
| AFLUX RF15 | 1,097 | 0.88 | 0.14 | 4 | 4,662 | 0.82 | 0.05 | 6 | 0.73 | 0.05 | 5 | 0.71 | 0.09 | 10 |
| ACLOUD | 3,431 | 0.7 | 0.06 | 2 | 16,747 | | | | 0.74 | 0.04 | 8 | | | |
| AFLUX | 3,690 | 0.81 | 0.17 | 4 | 11,652 | 0.81 | 0.08 | 6 | 0.75 | 0.07 | 5 | 0.74 | 0.08 | 9 |

The crucial hyperparameter of K-Means clustering is the total number of clusters. Therefore, we analyze the three heuristics distortion (Thorndike, 1953), Calinski–Habarasz index (Calinski and Harabasz, 1974), and silhouette score (Rousseeuw, 1987), which yield an optimal cluster number of four (Appendix B). However, not all clusters separate clearly due to transitional stages and inhomogeneous sea ice properties within MiRAC's footprint (Fig. B1b). Fish-eye images for all samples underline the high diversity in sea ice and snow properties (Fig. B2).

The occurrence of each cluster varies between the flights. Cluster 1 (C1) occurs more often than the other clusters with 52 % during RF08, C2 with 68 % during RF14, and C3 with 48 % during RF15. C4 occurs about 20 % of the time during RF08 and RF14, and 8 % of the time during RF15. It is unclear whether these changes relate to sea ice drift or temporal changes in ice properties due to the coarse temporal resolution and potential bias due to the flight pattern.

Each cluster shows distinct emissivity features (Fig. 5a). The lowest emissivity prevails in C1 and the highest in C4. C1 occurs over snow-covered sea ice (Fig. 5c) that might be classified as multiyear ice, which dominates during the AFLUX flights (Fig. A1). This also corresponds to the low skin temperature of 250 K for this cluster compared to the other clusters (Fig. 5b). Few open leads occur within C1 as water shows a spectral signature similar to this cluster. C4 occurs over nilas in refrozen leads (Fig. 5c). This agrees with generally warmer skin temperature compared to the other clusters (Fig. 5b). C4 is distinct from the other three clusters at 183, 243, and 340 GHz. C2 emissivities lie in between C1 and C4 at all frequencies. This cluster occurs over various surface types, predominantly over ice with visual properties of first-year ice. C3 emissivities lie close to C4 at 89 GHz and close to C1 at 243 and 340 GHz. This cluster occurs over young ice with more snow cover than the sea ice in C4. Hence, scattering within the upper snow layer could explain the lower emissivity at 243 and 340 GHz compared to C4. However, the emissivity is lower than in C2, where snow is also present, which indicates the importance of other factors such as snow density and grain size.



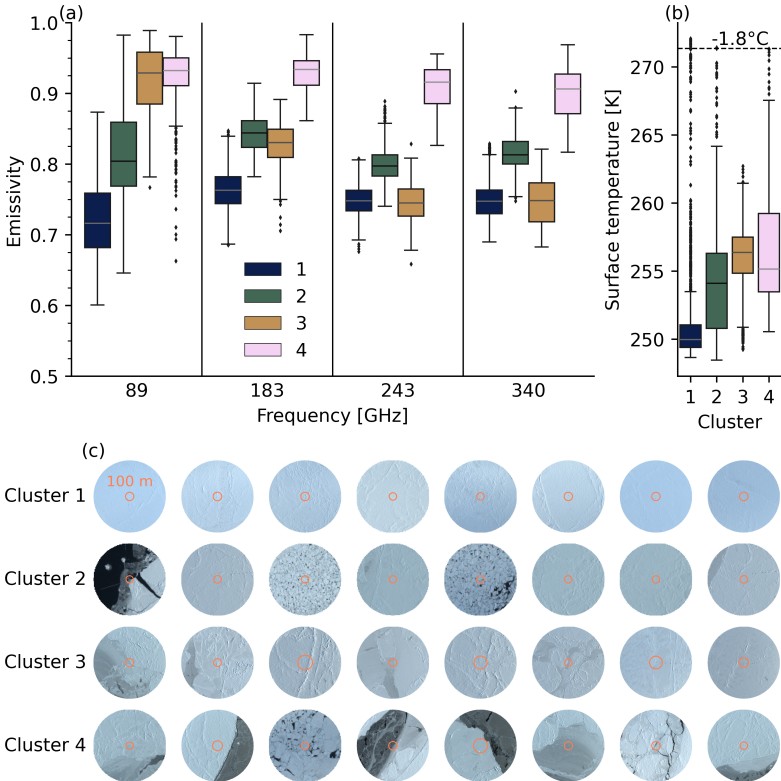

**Figure 5.** Comparison of sea ice emissivity and surface temperature across K-Means clusters. (a) Tukey boxplot depicting the distribution of sea ice emissivity at MiRAC frequencies within each K-Means cluster. (b) Tukey boxplot of the distribution of surface temperature within each K-Means cluster. (c) Fish-eye lens images representing the K-Means cluster centroids, i.e., for emissivity samples similar to the mean cluster emissivity, with a 100 m diameter nadir reference circle (see Fig. B2 for all images). It should be noted that the actual footprint might not lie within the indicated region due to the aircraft attitude making MiRAC-P point off-nadir by a few degrees and potential temporal shifts between the camera and MiRAC.

The evaluation of airborne emissivities reveals (1) low differences in median emissivity and interquartile range at 183, 243, and 340 GHz, (2) higher emissivities over nilas compared to multiyear ice at all frequencies, and (3) four distinct emissivity spectra. The similarity between 243 and 340 GHz implies a lower spectral sea ice emissivity variation in the submillimeter wave range. However, the emissivity variability at both frequencies is still significant and depends on the sea ice type, with the highest contrast between multiyear ice and nilas.



## 5 Comparison with satellites

### 5.1 Spatial variability at sub-footprint scale

The airborne and satellite observations resolve sea ice emissivity on different spatial scales. Hectometer-scale airborne observations resolve most leads, while kilometer-scale satellite observations partly smooth out these structures. Figure 6a showcases a *Polar 5* transect during AFLUX RF08, covering a 5 km lead mainly composed of nilas. MiRAC's 89 GHz emissivity exhibits a pronounced increase from multiyear or first-year ice to nilas and sharply decreases over a short section of open water. Consequently, emissivity clusters shift from C1 over multiyear or first-year ice to C4 over younger sea ice. The 5 km AMSR2

footprints partially resolve the lead, with higher emissivities over nilas, whereas the $16 \times 16$ km$^2$ MHS footprint cannot fully capture it. This example underscores the significance of sub-footprint-scale emissivity variations over spatially heterogeneous sea ice.

Next, we examine how emissivity varies with footprint size from 0.1 to 20 km from all airborne observations. We calculate the larger scale emissivity from mean airborne surface temperature and emission for each footprint size interval. The

interquartile range of the emissivity decreases rapidly with increasing footprint size during ACLOUD and AFLUX at all frequencies (Fig. 6b). For example, the variability of 100 m footprints at 340 GHz during AFLUX decreases by 42 % (65) at $5 \times 5$ km$^2$ ($16 \times 16$ km$^2$) footprint size. The lowest decrease occurs at 89 GHz during ACLOUD with 21 % (20 %) at $5 \times 5$ km$^2$ ($16 \times 16$ km$^2$) footprint size. Hence, the satellite footprint contains mean conditions where significant small-scale variability averages out.

### 5.2 Channel intercomparison


Before combining MiRAC with all satellite observations to study spectral variations on satellite scale up to 340 GHz, we must ensure that our collocation approach reproduces satellite observations at similar frequencies and observing geometries. The near-nadir (0 to 30°) 157 GHz MHS and 165.5 GHz ATMS channels are comparable to MiRAC's nadir 183 GHz channel. We compare these satellite channels instead of the 190.31 and 183.31±7 GHz channels due to their higher surface sensitivity and

lower uncertainty. Other channel or instrument combinations differ in incidence angle or polarization, making footprint-level comparisons less meaningful. For the comparison, we assume low emissivity gradients over the 157 to 183 GHz frequency range on satellite scale (Wang et al., 2017b).

Figure 7 illustrates the resampling process, transitioning from MiRAC's high-resolution emissivity to the satellite footprints, followed by the corresponding satellite emissivity and their difference for all AFLUX flights. Notably, MiRAC reveals

hectometer-scale emissivity features, such as leads, which MHS and ATMS miss due to their $16 \times 16$ km$^2$ footprint. This high hectometer-scale variability consistently occurs within each satellite footprint (Fig. 7, right column) and diminishes after resampling to the satellite footprint scale. The limited spatial coverage of MiRAC causes slightly higher emissivity variability compared to MHS and ATMS, as MiRAC only captures a narrow strip of the satellite footprint, e.g., during AFLUX RF08 near 80.4° N, 5° E (Fig. 7a) leading to the highest emissivity bias (Fig. 7d). However, the collocation method is robust for

most cases and yields MiRAC emissivities representative for the $16 \times 16$ km$^2$ satellite footprints. Moreover, the assessment





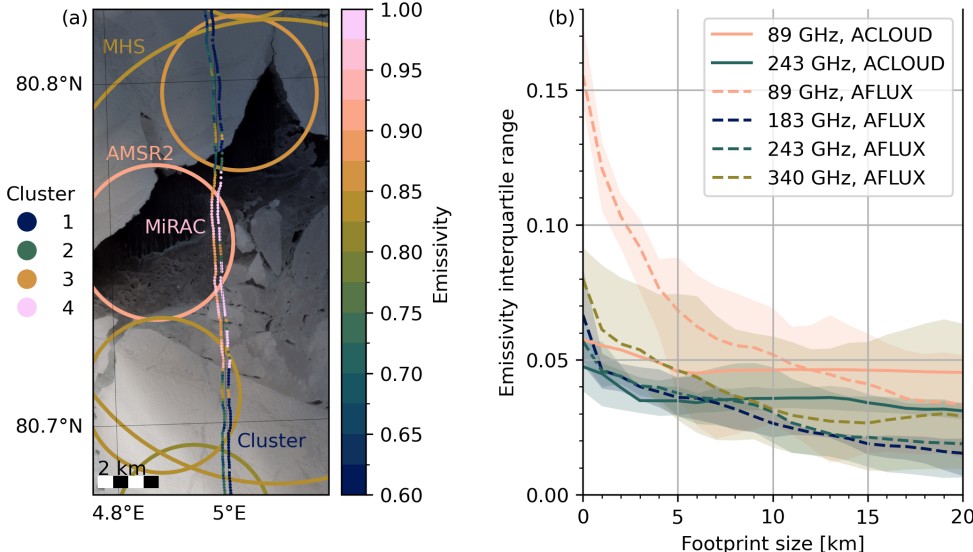

**Figure 6.** (a) Sea ice emissivity at 89 GHz from MiRAC (10:32 to 10:37 UTC), AMSR2 (11:02 UTC), and MHS/Metop-B (11:38 UTC) during AFLUX RF08 on 31 March 2019. The MiRAC emissivity cluster is displayed 100 m eastward of the emissivity. The actual MiRAC footprints lie in between the emissivity and emissivity cluster locations. The background is a Sentinel-2B L2A natural-color image at 14:37 UTC shifted 4 km northward to correct for sea ice drift. (b) Emissivity interquartile range as a function of footprint size from 0.1 to 20 km for all flights and channels. The spread represents the minimum and maximum interquartile range for each campaign.

of relative bias calculated as MHS or ATMS minus MiRAC emissivity divided by MiRAC emissivity yields insights into the consistency of MiRAC observations with satellites (Tables 4 and 5). This relative bias of -3 to 1 % lies well within MiRAC's 6 % uncertainty range at 183 GHz (see Table 3). The correlation between MiRAC and MHS or ATMS ranges from 0.4 to 0.6 and reflects the partial footprint overlap that reduces the representation of MiRAC for each satellite footprint. In summary, the comparison with MHS and ATMS provides confidence in the accuracy of our airborne emissivity estimates and the conversion from hectometer to satellite footprint scale. Hence, we can apply the same approach to other MiRAC channels up to 340 GHz.

## 5.3 Spectral variations

In this section, we collocate MiRAC with MHS and ATMS for near-nadir incidence angles from 0 to 30°, and SSMIS and AMSR2 to analyze spectral sea ice emissivity variations from 88 to 340 GHz and angular and polarization effects. We group all collocated emissivities by frequency, i.e., 88–92, 150–165 (only for satellites), 176–190, 243, and 340 GHz. The MiRAC observations are averaged to the footprints of each satellite instrument to ensure equal spatial sampling.

The channel-dependent emissivity variability observed on satellite scale during ACLOUD and AFLUX reveals distinct features related to spectral, angular, and polarization differences (Fig. 8). Low spectral differences occur during ACLOUD near nadir from 89 to 243 GHz (MHS and MiRAC) and at vertical polarisation from 91 to 150 GHz (SSMIS; Fig. 8a). As



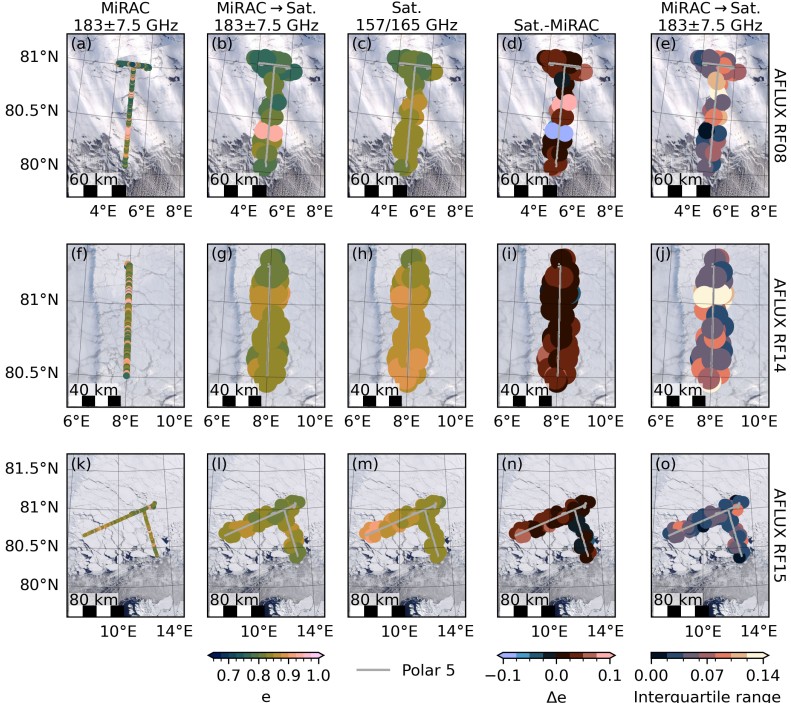

**Figure 7.** Comparison of emissivity from nadir 183 GHz MiRAC and near-nadir (0 to 30°) 157 GHz MHS and 165.5 GHz ATMS observations along the *Polar 5* flight track during AFLUX RF08, RF14, and RF15 (rows). (a, f, k) MiRAC emissivity at original resolution. (b, g, l) MiRAC emissivity resampled to satellite (Sat.) footprints. (c, h, m) satellite emissivity. (d, i, n) Emissivity difference between MiRAC and satellites (satellite minus MiRAC). (e, j, o) The MiRAC emissivity interquartile range within the satellite footprint. No 183 GHz observations from MiRAC were available during ACLOUD. The background images are MODIS/Terra composites of the same day (NASA Worldview). All footprints are approximated as circles. MiRAC's footprints are enlarged to 5 km diameter. The satellite footprint size corresponds to the footprint size at nadir.

expected, the 89 GHz emissivity shows a polarization signal of about 0.1. Hence, the purely Lambertian assumption in our emissivity calculation introduces an emissivity bias less than 2 % for MiRAC. The combination of both polarizations from SSMIS or AMSR2 to quasi-vertical polarization following Eq. (1) reduces the absolute emissivity difference to the 0 to 30° emissivity of MHS. Furthermore, the horizontally polarized 89 GHz channel of MiRAC lies closer to the horizontally polarized channels of SSMIS and AMSR2. Spectral differences during AFLUX exceed those of ACLOUD, which might be explained by

the contrasting sea ice properties with melting conditions during ACLOUD and much colder and dryer sea ice and snow during AFLUX (Fig. 8b). The near-nadir emissivity remains constant from 89 to 183 GHz and decreases toward 243 and 340 GHz. No significant spectral emissivity difference can be detected in the 165 to 183 GHz frequency range where all satellites fall within MiRAC's 6 % uncertainty (see Table 3). The decrease toward 243 GHz exceeds the 243 GHz emissivity uncertainty. The AFLUX emissivities show a lower polarization difference at 89 GHz than ACLOUD, which can be explained by the lower





**Table 4.** Comparison of collocated emissivity from nadir 183 GHz MiRAC and near-nadir (0 to 30°) 157 GHz MHS observations during the three AFLUX flights. Values denote the number of collocated satellite footprints (Count), median, interquartile range (IQR), relative bias (Rel. bias), i.e., MHS emissivity minus MiRAC emissivity divided by MiRAC emissivity, relative root-mean-square deviation (Rel. RMSD) normalized by MiRAC emissivity, and Pearson's correlation coefficient (Corr.).

| Campaign/RF | Count | Median | | IQR | | Rel. bias | Rel. RMSD | Corr. |
|---|---|---|---|---|---|---|---|---|
| | | MiRAC | MHS | MiRAC | MHS | [%] | [%] | |
| AFLUX RF08 | 36 | 0.79 | 0.81 | 0.04 | 0.04 | 1 | 5 | 0.56 |
| AFLUX RF14 | 34 | 0.84 | 0.83 | 0.02 | 0.02 | -1 | 2 | 0.48 |
| AFLUX RF15 | 68 | 0.83 | 0.82 | 0.02 | 0.02 | -1 | 2 | 0.5 |

**Table 5.** Comparison of collocated emissivity from nadir 183 GHz MiRAC and near-nadir (0 to 30°) 165.5 GHz ATMS observations during the three AFLUX flights. The columns are identical to Table 4.

| Campaign/RF | Count | Median | | IQR | | Rel. bias | Rel. RMSD | Corr. |
|---|---|---|---|---|---|---|---|---|
| | | MiRAC | ATMS | MiRAC | ATMS | [%] | [%] | |
| AFLUX RF08 | 18 | 0.81 | 0.81 | 0.04 | 0.04 | 1 | 4 | 0.63 |
| AFLUX RF14 | 13 | 0.84 | 0.82 | 0.03 | 0.01 | -3 | 4 | 0.58 |
| AFLUX RF15 | 15 | 0.83 | 0.82 | 0.02 | 0.01 | -1 | 3 | 0.42 |

amount of open water between ice floes during AFLUX. The emissivity of the 89 GHz MiRAC channel lies in between the horizontally polarized AMSR2 and SSMIS and the near-nadir MHS and ATMS channels.

Different instruments show similar emissivity distributions at similar channels. For example, the three MHS and ATMS channels exhibit nearly identical distributions during AFLUX (see Fig. 8b). Additionally, the polarized 89 GHz channels of SSMIS and AMSR2 show good agreement. However, during ACLOUD, emissivity differences between AMSR2 and SSMIS

are noted for the vertically polarized channel, primarily due to the low number of collocated AMSR2 footprints with MiRAC. Additionally, AMSR2 shows higher variability due to its smaller footprint than SSMIS.

Furthermore, MiRAC distributions align with MHS and ATMS near nadir. The increased emissivity variability of MiRAC's 25° inclined 89 GHz channel, compared to MHS and ATMS, may be explained by its horizontal polarization. When comparing the vertically and horizontally polarized SSMIS and AMSR2 channels, the horizontal polarization exhibits higher variability,

consistent with findings from experiments by Shokr et al. (2009).

The consistent outcomes from spaceborne and airborne observations unveil a first-time representation of sea ice emissivity variability from 89 to 340 GHz. As detected by MiRAC, hectometer-scale emissivity variations smooth out when observed from a satellite perspective. Our analysis shows a potential decline in emissivity from 183 to 243 GHz under cold and dry




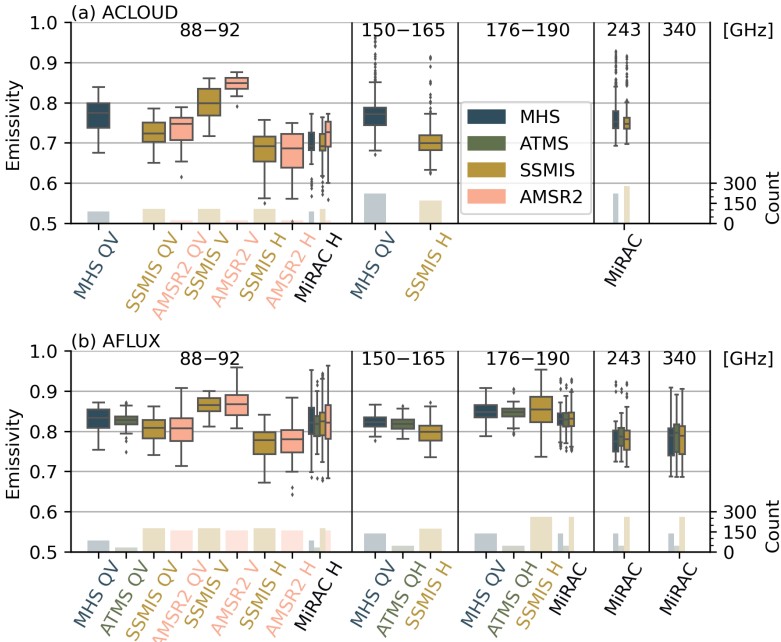

**Figure 8.** Tukey boxplots of the collocated emissivity during (a) ACLOUD and (b) AFLUX for the frequency ranges 88–92, 150–165, 176–190, 243, and 340 GHz derived from MHS (0 to 30°), ATMS (0 to 30°), SSMIS (53°), AMSR2 (55°), and MiRAC (25° at 89 GHz and 0° at 183, 243, and 340 GHz). The secondary axis denotes the count of collocated footprints. Quasi-vertical polarization in SSMIS QV and AMSR2 QV is characterized by a dominant 64 and 67 % influence of horizontal polarization, respectively. The 88 to 92 GHz satellite footprint count might be lower than above 150 GHz because satellite footprints are excluded if the nearest MiRAC channel contains no emissivity.

conditions during AFLUX. This spectral pattern occurs within airborne emissivity clusters C3 (young ice) and to some extent
in C2 (first-year ice) and C4 (nilas), but notably not in C1 (multiyear ice), and prevails after resampling onto satellite scale. These cluster differences underscore the importance of the spatial distributions of sea ice types.

## 6 Conclusions

The upcoming launches of ICI and EPS Sterna with novel frequencies above 200 GHz and AMSR3 with novel AMSR2-like resolution at 183 GHz require an improved understanding of the sea ice emissivity to separate atmospheric and surface
microwave signals under dry polar conditions (Wang et al., 2017b). However, few field observations measured sea ice emissivity at such high frequencies at hectometer scale resolution. Therefore, we analyzed sea ice emissivity variations observed with the microwave radiometer MiRAC during the airborne field campaigns ACLOUD (summer 2017) and AFLUX (spring 2019). The flights analyzed here covered about 1,700 km flight distance; 7,000 samples were collected at 89 GHz (25° incidence angle, H-pol); 28,000 samples at 243 GHz (nadir); and 11,000 samples at 183 and 340 GHz (nadir).



Our first objective was to identify critical physical sea ice and snow properties affecting emissivity up to submillimeter waves. The sea ice emissivity exhibits a high variability from about 0.65 to 1, with the lowest emissivities at 89 GHz. The 89 GHz distribution showed higher variability than the nadir channels due to its inclination and horizontal polarization. MiRAC resolves sea ice emissivity features that align with sea ice and snow properties identified from visual imagery. Four emissivity spectra from 89 to 340 GHz could be identified through K-Means clustering. They occur predominantly over multiyear, first-year,

young ice, and nilas. However, the emissivity variability for each cluster is significant due to variations in snow or sea ice microphysical properties and mixed types within the radiometer footprint. The lowest emissivity occurs over multiyear ice and the highest emissivity over nilas as found in previous studies at 89 and 183 GHz (NORSEX Group, 1983; Hewison and English, 1999; Hewison et al., 2002).

Our second objective was to relate the observed hectometer-scale emissivity observations to the satellite scale. We collo-

cated MiRAC with MHS, ATMS, SSMIS, and AMSR2 for this. Satellite instruments do not resolve hectometer-scale sea ice emissivity variations observed by MiRAC due to their larger footprints. By averaging the airborne observations, we estimated the decrease in the emissivity interquartile range with increasing footprint size. The reduction in interquartile range is most significant during AFLUX when leads induce significant hectometer-scale emissivity variations. For example, the emissivity interquartile range decreases by almost half from the hectometer scale to a footprint of $16 \times 16 \,\mathrm{km}^2$ typical for microwave satel-

lite instruments. We find high agreement between MHS and ATMS and MiRAC emissivities near 183 GHz. During AFLUX, the emissivity decreases significantly from 183 to 243 GHz, while they remain almost constant during ACLOUD. The estimates provided here might represent emissivities that future satellites such as ICI and EPS Sterna observe.

The study's implications are as follows:

– Hectometer-scale frequency dependency: The 183, 243, and 340 GHz channels exhibit similar hectometer-scale sea ice

emissivity variations at nadir independent of sea ice type, e.g., multiyear ice and nilas. This finding is crucial for the development of airborne retrieval methods.

– Spatial and temporal representation: At the satellite footprint scale, hectometer-scale sea ice emissivity variations average out, which facilitates sea ice emissivity parameterization. However, these variations become relevant for higher-resolution channels, such as AMSR2.

– Emissivity frequency extrapolation: The relatively low spectral emissivity variation from 89 to 340 GHz at the satellite scale at nadir supports the first-order approximation of constant emissivities over sea ice within existing parameterizations such as TELSEM$^2$ (Wang et al., 2017b). Accounting for spatial and temporal emissivity variations appears more relevant than spectral gradients.

This study has several limitations:

– Channel intercomparison: The 25° inclination and horizontal polarization of the 89 GHz channel affects the comparison with the 183 to 340 GHz nadir-viewing channels by likely increasing its variability and lowering its emissivity compared





to an 89 GHz nadir-viewing channel. A quantification of this effect might be possible from the airborne HALO–(AC)[3] campaign in spring 2022 (Wendisch et al., 2021).

- Surface temperature assumption: Using the surface skin temperature instead of the emitting layer temperature imposes a frequency-dependent bias on the emissivity during AFLUX.

- Sea ice and snow properties: The aerial images provide only a broad perspective on sea ice and snow properties, with limitations in providing any (vertical profiles of) sea ice microphysics, such as density, grain size, or salinity.

- Spatial resolution: MiRAC's hectometer scale may not capture smaller sea ice features such as ridges or melt ponds, which could influence emissivity.

- Spatial and temporal limitation: Field observations are limited in space (approximately 100 km) and time (five days), potentially restricting the generalizability of findings across polar regions.

Three primary challenges persist in comprehending sea ice emissivity variations to advance atmospheric and surface retrievals over sea ice. First, the relatively unexplored emissivity dependence on polarization and incidence angle, especially at frequencies above 200 GHz, demands comprehensive investigation. To address this, potential solutions include utilizing shipborne or airborne observations with scanning radiometers. Second, the high uncertainty due to atmospheric emissions that mask spectral emissivity features, particularly at 340 GHz and over the more reflective multiyear ice, requires the simultaneous measurement of near-surface downwelling atmospheric TB for emissivity calculation. Third, observed emissivity spectra must be combined with sea ice and snow microphysics in situ measurements to advance radiative transfer modeling. In summary, these challenges can help address remaining knowledge gaps in sea ice microwave emissivity, with implications for current and upcoming satellite missions. Future work will focus on separating sea ice and atmospheric signals under all-sky conditions.

*Code and data availability.* The code for this study and a usage example of the published emissivity data is available on Zenodo at https://doi.org/10.5281/zenodo.10533864 (Risse, 2024). The MiRAC emissivity data is currently accessible upon request and will be made publicly available on PANGAEA. MiRAC-A measurements during ACLOUD were obtained from https://doi.pangaea.de/10.1594/PANGAEA.899565 (Kliesch and Mech, 2019) and during AFLUX from https://doi.org/10.1594/PANGAEA.944506 (Mech et al., 2022b). MiRAC-P measurements during ACLOUD were obtained from https://doi.org/10.1594/PANGAEA.944070 (Mech et al., 2022c) and during AFLUX from https://doi.org/10.1594/PANGAEA.944057 (Mech et al., 2022d). Camera images during AFLUX were obtained from https://doi.org/10.1594/PANGAEA.901024 (Jäkel et al., 2021). KT-19 measurements during ACLOUD were obtained from https://doi.org/10.1594/PANGAEA.900442 (Stapf et al., 2019) and during AFLUX from https://doi.org/10.1594/PANGAEA.932020 (Stapf et al., 2021). Dropsonde measurements during ACLOUD were obtained from https://doi.org/10.1594/PANGAEA.900204 (Ehrlich et al., 2019a) and during AFLUX from https://doi.org/10.1594/PANGAEA.922004 (Becker et al., 2020). Nose boom measurements during ACLOUD were obtained from https://doi.org/10.1594/PANGAEA.902849 (Hartmann et al., 2019) and during AFLUX from https://doi.org/10.1594/PANGAEA.945844 (Lüpkes et al., 2022). Aircraft position and orientation were obtained from the ac3airborne intake catalog (Mech et al., 2022e). Radiosoundings in Ny-Ålesund were obtained from https://doi.org/10.1594/PANGAEA.914973 (Maturilli, 2020). The sea–land mask for Svalbard



was obtained from the Kartdata Svalbard 1:100 000 (S100 Kartdata) / Map Data of the Norwegian Polar Institute at https://doi.org/10.
21334/npolar.2014.645336c7 (Norwegian Polar Institute, 2014). The SSMIS/DMSP-F16 L1C TB data were obtained from https://doi.org/
10.5067/GPM/SSMIS/F16/1C/07 (Berg, 2021a). The SSMIS/DMSP-F17 L1C TB data were obtained from https://doi.org/10.5067/GPM/
SSMIS/F17/1C/07 (Berg, 2021b). The SSMIS/DMSP-F18 L1C TB data were obtained from https://doi.org/10.5067/GPM/SSMIS/F18/1C/07
(Berg, 2021c). The AMSR2/GCOM-W1 L1C TB data were obtained from https://doi.org/10.5067/GPM/AMSR2/GCOMW1/1C/07 (Berg,
2022a). The MHS/Metop-A L1C TB data were obtained from https://doi.org/10.5067/GPM/MHS/METOPA/1C/07 (Berg, 2022b). The
MHS/Metop-B L1C TB data were obtained from https://doi.org/10.5067/GPM/MHS/METOPB/1C/07 (Berg, 2022c). The MHS/Metop-
C L1C TB data were obtained from https://doi.org/10.5067/GPM/MHS/METOPC/1C/07 (Berg, 2022d). The MHS/NOAA-18 L1C TB
data were obtained from https://doi.org/10.5067/GPM/MHS/NOAA18/1C/07 (Berg, 2022e). The MHS/NOAA-19 L1C TB data were ob-
tained from https://doi.org/10.5067/GPM/MHS/NOAA19/1C/07 (Berg, 2022f). The ATMS/SNPP L1C TB data were obtained from https:
//doi.org/10.5067/GPM/ATMS/NPP/1C/07 (Berg, 2022g). The ATMS/NOAA-20 L1C TB data were obtained from https://doi.org/10.5067/
GPM/ATMS/NOAA20/1C/05 (Berg, 2022h). The NE23 Level 4 Arctic sea and ice surface temperatures were obtained from https://doi.
org/10.48670/moi-00123 (Nielsen-Englyst et al., 2023). The AMSR2 sea ice concentration data by the University of Bremen was retrieved
from https://data.seaice.uni-bremen.de/ (Spreen et al., 2008). The AMSR2/ASCAT multiyear ice concentration data by the University of
Bremen was retrieved from https://seaice.uni-bremen.de/multiyear-ice/data-access/ (Melsheimer and Spreen, 2022). Sentinel-2B L2A im-
ages were obtained from the Copernicus Data Space Environment https://doi.org/10.5270/S2_-znk9xsj (European Space Agency, 2021).
MODIS/Terra images were retrieved from the NASA Worldview application at https://worldview.earthdata.nasa.gov. The satellite bandpass
information was obtained from the EUMETSAT Numerical Weather Prediction Satellite Application Facility at https://nwp-saf.eumetsat.int/
site/software/rttov/download/coefficients/spectral-response-functions/ (NWP SAF, 2024).

## Appendix A: Multiyear ice concentration maps

We include maps of multiyear ice concentration for additional context to the three AFLUX flights (Fig. A1). The multiyear
ice concentration lies mainly around 50 % with higher concentrations in the northern parts of RF08 and RF14. The case study
transect during RF08 lies within a pixel of about 75 % multiyear ice concentration (Fig. A1a).

## Appendix B: Optimal number of K-Means emissivity clusters

The K-Means algorithm assigns a cluster to each normalized emissivity spectrum across the four MiRAC frequencies. The
normalization subtracts the mean and scales the emissivity of each channel to unit variance, which ensures equal weighting
between the four channels. However, the absolute number of clusters ($K$) is unknown and needs to be defined objectively.
Therefore, we evaluate three metrics for cluster sizes between two and ten to identify the optimal $K$ (Fig. B1a). The distortion
represents the sum of squared distances across all samples to their assigned cluster centroid (Thorndike, 1953). The distortion
ideally follows an elbow-shaped curve with a decrease until the optimal $K$ value and constant distortion for higher $K$ values.
The distortion curve of the emissivity samples flattens slightly after a $K$ value of four. The Calinski–Habarasz index determines
the ratio of the sum of between-cluster dispersion and within-cluster dispersion, i.e., separation and cohesion (Calinski and
Harabasz, 1974). Higher Calinski–Habarasz index values correspond to optimal clustering with well-separated and dense



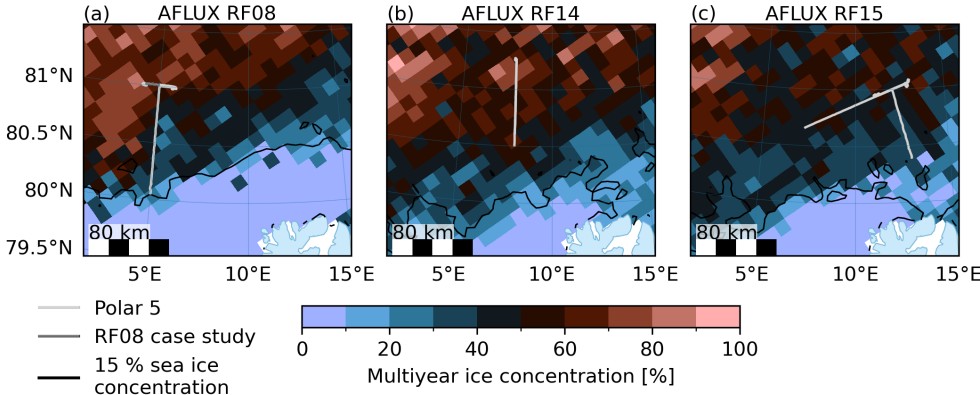

**Figure A1.** Maps of the *Polar 5* flight track, sea ice edge indicated as 15 % sea ice concentration isoline (Spreen et al., 2008), and multiyear ice concentration (Melsheimer and Spreen, 2022) during (a) AFLUX RF08 including the case study transect, (b) RF14, and (c) RF15.

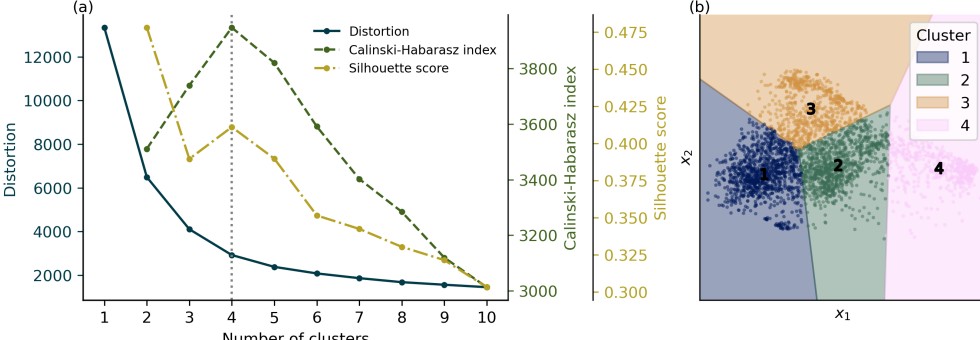

**Figure B1.** (a) The K-Means clustering metrics distortion, Calinski–Habarasz index, and silhouette score as a function of number of clusters. (b) Clustered emissivity spectra projected along the first two principal components ($x_1$, $x_2$) using K-Means. The K-Means cluster boundaries are approximated as a Voronoi diagram based on the cluster centroid projection. The cluster numbers are located at the centroid position.

clusters. The index peaks at a $K$ value of four and decreases for higher and lower values (Fig. B1a). The silhouette score represents the mean silhouette coefficient, i.e., the similarity of a sample to its own compared to other clusters (Rousseeuw, 1987). Silhouette coefficients of 1 (-1) indicate correct (wrong) class assignment. On average, the silhouette score is 0.37 for two to ten clusters. The silhouette score is highest for two clusters and shows a secondary peak for four. All three metrics indicate that the emissivity spectra divide optimally into four clusters. The two-dimensional principal component analysis (Hotelling, 1933) compression shows the four identified emissivity clusters (Fig. B1b). Overall, the emissivity clusters are well-separated with gradual transitions due to mixed types within the radiometer's footprint or transitional stages of the sea ice. Fish-eye lens images resolve these mixed types and transitional stages (Fig. B2).




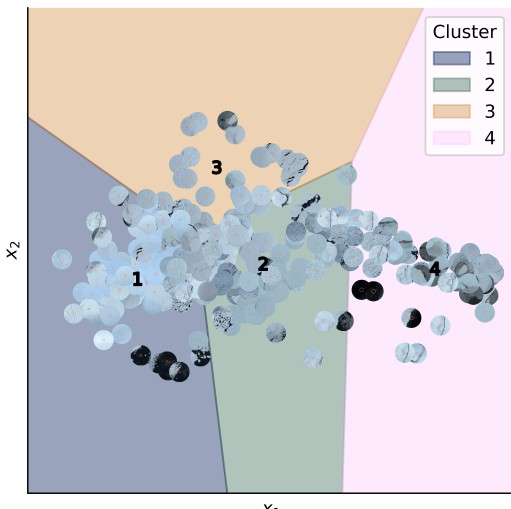

**Figure B2.** Fish-eye lens images corresponding to the emissivity samples in Fig. B1b. The K-Means cluster boundaries are approximated as a Voronoi diagram based on the cluster centroid projection onto the first two principal components ($x_1$, $x_2$). The cluster numbers are located at the centroid position.

*Author contributions.* NR conducted the emissivity retrieval, data analysis, and visualization and prepared the manuscript. SC, MM, and NR conceptualized the study. SC and MM carried out the field observations. CP and GS provided valuable expertise in interpreting emissivity signatures. All authors reviewed and edited the manuscript.

*Competing interests.* The authors declare that they have no conflict of interest.

*Acknowledgements.* We gratefully acknowledge the funding by the German Research Foundation [Deutsche Forschungsgemeinschaft (DFG)] of the Transregional Collaborative Research Center SFB/TRR 172 "Arctic Amplification: Climate Relevant Atmospheric and Surface Processes, and Feedback Mechanisms (AC)[3]" (Project-ID 268020496). We sincerely thank the Alfred Wegener Institute for providing and operating the *Polar 5* aircraft, and our sincere appreciation extends to the dedicated crew and technicians who supported its missions. We acknowledge the use of imagery from the NASA Worldview application (https://worldview.earthdata.nasa.gov), part of the NASA Earth Observing System Data and Information System (EOSDIS). Furthermore, we acknowledge the freely available Python packages, including but not limited to numpy (Harris et al., 2020), pandas (McKinney, 2010), xarray (Hoyer and Hamman, 2017), scipy (Virtanen et al., 2020), and gdal (Warmerdam, 2008) for data analysis, matplotlib (Hunter, 2007), seaborn (Waskom, 2021), and cartopy (UK Met Office, 2023) for visualization, and scikit-learn (Pedregosa et al., 2011) for K-Means clustering. We sincerely appreciate Fabio Crameri for providing scientific colormaps via an open repository, enhancing the visual quality of this work (Crameri, 2018). We acknowledge the use of OpenAI's



language models, including the Generative Pre-trained Transformer 3.5 (GPT-3.5) via ChatGPT and GPT-4 via GitHub Copilot, in preparing

and refining written content and code, respectively.



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
