# Peer review of "Assessing the sea ice microwave emissivity up to submillimeter waves from airborne and satellite observations"

_EGUsphere, 2024_

## Author Comment (AC1)

**Authors' Response to Reviews of**

**Assessing the sea ice microwave emissivity up to submillimeter waves from airborne and satellite observations**

Nils Risse, Mario Mech, Catherine Prigent, Gunnar Spreen, and Susanne Crewell
*The Cryosphere,* https://doi.org/10.5194/egusphere-2024-179

RC: *Reviewers' Comment*,     AR: Authors' Response,     □ Manuscript Text

**1. RC1, Dr. Tim Hewison**

**1.1. General comment**

RC: *The manuscript presents a valuable analysis of sea ice emissivity measurements, including novel observations at submillimeter wavelengths, which will be of considerable interest to the remote sensing community, given the forthcoming AWS, ICI and Sterna satellite missions. Importantly it relates the variability on the different scales observed by airborne and satellite sensors. The authors could consider further analysis to include a quantification of the scene variability on other scales - e.g. through the use of variograms/structure functions. This could extend the application of the results to other applications. It is generally well-written and the conclusions, in particular, are clear. However, there are several cases where key details of the methodology are missing, which would make it very difficult to reproduce the results. Examples are given below. Furthermore, more attention needs to be paid to uncertainties - especially those introduced by various assumptions (see below).*

AR: The authors would like to thank Dr. Tim Hewison for providing highly valuable and constructive feedback on this manuscript. We have carefully considered all the comments and provided responses below. The revised version provides further details on the methodology to facilitate the reproducibility of the results. Referee comments are given in bold text.

AR: Variograms (or semi-variograms; e.g., Mälicke et al., 2020) or structure functions (e.g., Kitchen, 1989) describe the spatial dependence as a function of separating distance. Variograms are thus conducted with data of similar spatial resolution, e.g., MiRAC or satellites. They provide the basis for spatial interpolation, which we did not aim for in this work. However, we agree that the parameters of a variogram (nugget, sill, effective range) and their dependence on environmental conditions (e.g. for soil moisture measurement networks; Lakhankar et al., 2010) or frequency in our case could provide new information, but such an analysis was out of the scope of this study. Instead, we focused on the emissivity variability as a function of footprint size. This allows us to bridge from small-scale airborne observations to large-scale satellite observations. Hence, it also provides a representation of the underlying spatial covariance described by a variogram. A key difference is that this approach does not use a distance but a radius, and we do not quantify the deviations from the central location but average them and compute the variability over all samples. Hence, we change spatial resolution with increasing radius that can be compared to satellite observations, e.g., 5 or 16 km. However, to allow further analysis of the data using other geostatistical methods, such as variograms, we provide the derived emissivities on PANGAEA.

Kitchen, M. (1989), Representativeness errors for radiosonde observations. Q.J.R. Meteorol. Soc., 115: 673-700. https://doi.org/10.1002/qj.49711548713

Lakhankar T, Jones AS, Combs CL, Sengupta M, Vonder Haar TH, Khanbilvardi R. Analysis of large scale spatial variability of soil moisture using a geostatistical method. Sensors (Basel). 2010;10(1):913-32. doi: 10.3390/s100100913. Epub 2010 Jan 25. PMID: 22315576; PMCID: PMC3270877.

Mälicke, M., Hassler, S. K., Blume, T., Weiler, M., and Zehe, E.: Soil moisture: variable in space but redundant in time, Hydrol. Earth Syst. Sci., 24, 2633–2653, https://doi.org/10.5194/hess-24-2633-2020, 2020.

**1.2. Line 106**

**RC:** *Please provide a reference to a document describing the calibration and bias correction procedures.*

AR: We added two references to other work that describe the calibration and bias correction procedures. The reference Mech et al. (2019) describes the calibration procedure, including internal calibration and absolute calibration. The bias correction procedure follows the approach that Konow et al. (2019) applied to airborne observations of the HAMP radiometer onboard the HALO research aircraft, which we transfer to the Polar 5 aircraft.

Konow, H., Jacob, M., Ament, F., Crewell, S., Ewald, F., Hagen, M., Hirsch, L., Jansen, F., Mech, M., and Stevens, B.: A unified data set of airborne cloud remote sensing using the HALO Microwave Package (HAMP), Earth Syst. Sci. Data, 11, 921–934, https://doi.org/10.5194/essd-11-921-2019, 2019.

> The instrument receivers were calibrated with a two-point calibration using liquid nitrogen and an internal target at the beginning of each campaign. In addition, MiRAC-A performed gain calibrations every 15 min, and MiRAC-P every 20 min during flights using an internal target (Mech et al., 2019). After the campaign, we applied a bias correction of the 89 GHz TBs following Konow et al. (2019) based on Passive and Active Microwave Radiative Transfer (PAMTRA; Mech et al., 2020) forward simulations by using dropsonde profiles under clear-sky conditions over the open ocean extended by ERA5 reanalysis (Hersbach et al., 2020) to the top of the atmosphere and a sea surface temperature analysis (UK Met Office, 2012) as input.

**1.3. Line 116**

**RC:** *Does this uncertainty combine systematic and random effects? It is important that they can be treated separately in evaluating the uncertainty of the emissivity estimates on different spatial scales.*

AR: Yes, the uncertainty of the measured TB combines systematic and random effects. The random noise used in our work is an upper bound. We estimate it from 20 consecutive 30 s measurement intervals over open ocean under clear sky during ACLOUD RF10. The TB standard deviation averaged over all 20 intervals ranges from 0.2 to 0.3 K depending on the channel. The bias correction is performed with a limited number of clear-sky dropsonde observations that have a random and systematic uncertainty (relative humidity, temperature, surface temperature, surface wind speed, forward simulation, ...). Therefore, the reference itself is very likely biased, which we account for by increasing the 89 GHz TB error by 2 K. In our study, we treat both components of the TB error as systematic and do not separate them when averaging multiple observations. Generally, it is correct that random TB noise would cancel out when averaging multiple observations. Our approach aims at reducing the complexity of the emissivity error estimation while still providing a robust quantitative estimate. A separation of random and systematic errors would have a negligible effect on the emissivity error because of the much smaller contribution of random TB errors compared to systematic error sources that affect the emissivity, i.e., surface temperature, atmospheric temperature, relative humidity, and TB bias correction (89 GHz channel). Our final uncertainty represents a confidence interval dominated by systematic

uncertainty. We will add this information and a short discussion to the manuscript.

> The TB noise is about 0.5 K for MiRAC-A (Küchler et al., 2017) and -P (Mech et al., 2019), which is an upper bound of the observed TB noise of 0.2 to 0.3 K depending on the channel from a homogeneous time series during ACLOUD RF10. This random noise cancels out when averaging, but we do not consider this here as systematic effects dominate the overall emissivity uncertainty (see Sect. 3.2). Hence, we assume the overall TB uncertainty from bias correction and noise to be 2.5 K at 89 GHz and 0.5 K at all other frequencies.

**1.4. Line 153**

**RC:** *What uncertainty is added by this assumption? Could it introduce significant biases? E.g. during strong surface inversions - in this case, would it be better to assume a linear change from the flight level to the surface?*

**AR:** We now added information on the lowest flight altitude to the paragraph to clarify that it is just about 100 m and not 3 km above ground. The quantification of emissivity uncertainty due to atmospheric temperature and humidity profiles described in the manuscript also includes the near-surface atmosphere. Therefore, it also includes extrapolation uncertainties close to the ground.

Tjernström, M. and Graversen, R.G. (2009), The vertical structure of the lower Arctic troposphere analysed from observations and the ERA-40 reanalysis. Q.J.R. Meteorol. Soc., 135: 431-443. doi: 10.1002/qj.380

> We assume constant temperature and humidity from the lowest flight altitude of about 100 m down to the surface if no dropsonde information is available over sea ice. The air temperature measured at these heights differs less than 5 K from the mean surface temperature, which indicates that the profiles capture typical Arctic surface temperature inversions (e.g., Tjernström and Graversen, 2009) .

**1.5. Line 158**

**RC:** *What infrared emissivity is assumed to open water? Is this a function of sea state?*

**AR:** We assume an infrared emissivity of 0.995 for both sea ice and water, close to the constant value of 0.996 in Hoyer et al. (2017) and Thielke et al. (2022) for mixed sea ice and ocean surfaces. The emissivity of 0.995 represents the observed infrared emissivity of various ice types and sea water within the KT-19 band pass from 9.6 to 11.5 $\mu$m (Hori et al., 2006). We do not vary the IR emissivity of sea water along the flight track due to discontinuous information on the presence of water within the radiometer footprint and a lack of accurate sea water state description. This approach aligns with previous studies on sea ice emissivity that observed mixtures of open water and sea ice (e.g., Hewison and English, 1999). We clarified the approach in the revised manuscript.

Høyer, J. L., Lang, A. M., Tonboe, R., Eastwood, S., Wimmer, W., and Dybkjær, G. Report from Field Inter-Comparison Experiment (FICE) for ice surface temperature. Danish Meteorological Institute (2017).

Thielke, L., Huntemann, M., Hendricks, S., Jutila, A., Ricker, R., and Spreen, G. (2022). Sea ice surface temperatures from helicopter-borne thermal infrared imaging during the MOSAiC expedition. Sci. Data, 9, 364. doi:10.1038/s41597-022-01461-9

> The infrared TB is converted to surface skin temperatures with an infrared emissivity of 0.995 similar to Hoyer et al. (2017) and Thielke et al. (2022), which approximates the infrared emissivity of typical sea ice types and ocean with an accuracy of 0.01 to 0.02 (Hori et al., 2006).

**1.6. Line 163**

**RC:** *Is this bias compensated for? How is it accounted for in the uncertainty analysis?*

**AR:** The bias between KT-19 and the thermal infrared-based satellite reanalysis by Nielsen-Englyst et al. (2023) is not compensated for or directly considered in the uncertainty analysis. We did not apply an offset to NE23 to match KT-19. This results in a potential overestimation of the 89 GHz emissivities from satellites compared to MiRAC during ACLOUD, where NE23 temperatures are 4 to 6 K lower than KT-19. However, this offset would not lead to a relevant change of the spectral variability that we observe. The emissivity change can be estimated directly from our uncertainty analysis. The surface temperature error of $\pm 3$ K for ACLOUD results in 2 % emissivity uncertainty. If we multiply this by a factor of 2, we find that the offset between MiRAC and satellites is about 4 % or 0.03 in emissivity. This is equivalent to the 89 GHz uncertainty during AFLUX for MiRAC. Therefore, we do not shift the NE23 estimates to match KT-19. We included the bias between NE23 and KT-19 in the detailed discussion of the spectral emissivity variations in Sect. 5.3

> Low spectral differences occur during ACLOUD near nadir from 89 to 243 GHz (MHS and MiRAC) and at vertical polarisation from 91 to 150 GHz (SSMIS; Fig. 8a). The higher satellite emissivity can be explained by the underestimation of the NE23 skin temperature compared to KT-19.

**1.7. Line 165**

**RC:** *(1) It would be helpful to include a short summary of how the NE23 analysis is derived. (2) It should also be clarified whether the NE23 analysis is used instead of the KT-19 measurements for emissivity calculations airborne as well as satellite measurements, and, if so, why, and (3) how this contributes to the overall uncertainty - especially in the context of surface temperature gradients through sea ice and any overlying snow layers.*

**AR:** Response to (1): The NE23 product obtains daily sea and sea ice surface temperatures from clear-sky thermal infrared satellite observations derived by optimal interpolation. The reference contains further details on the included sensors. During ACLOUD and AFLUX, the product uses the operational OSISAF IST product (OSI-205; Dybkjaer et al., 2018) based on 11 and 12$\mu$m AVHRR observations and an AVHRR-based cloud mask. Sea ice concentration fields from OSISAF are used to separate sea and sea ice temperatures. We summarized this in the revised manuscript.

Dybkjaer, G., Eastwood, S., Borg, A.L., Høyer, J.L., Tonboe, R., 2018. Algorithm theoretical basis document (ATBD) for the OSI SAF Sea and sea ice surface temperature L2 processing chain. OSI205a and OSI205b.

**AR:** Response to (2): The NE23 is only used for satellite, not airborne observations. We clarified this in Sect. 2.4. This is a natural choice as KT-19 measures simultaneously to MiRAC at a comparable spatial resolution. On the other hand, KT-19 is not used for satellites because the NE23 product observes the sea ice at a similar spatial resolution (about 5 km) as the satellites and provides better spatial coverage than KT-19. We will mention this in the revised manuscript.

**AR:** Response to (3): Using two different data sets comes at a cost of biases between both propagating into the emissivity estimates. Lower temperature biases occur during AFLUX (similar to 1.5 K found for KT-19

observations during IceBridge flights described in the reference publication), and higher temperature biases occur during ACLOUD. These higher biases during ACLOUD affect the comparison of satellite and airborne emissivities shown in Fig. 8. The satellite emissivities might be higher than the airborne emissivities due to the lower NE23 surface temperature (see the reply on line 163 and update of the manuscript). Generally, we expect low systematic differences between both estimates because they rely on the same measurement principle. They are not affected by surface temperature gradients through sea ice and any overlying snow layers because they measure only the upper few millimeters of the snow or ice (Warren, 1982).

Warren, S. G. (1982), Optical properties of snow, Rev. Geophys., 20(1), 67–89, doi: 10.1029/RG020i001p00067.

> We use KT-19 as input to the sea ice emissivity calculation for MiRAC.We also require an accurate description of the surface temperature at the satellite footprint scale with higher spatial coverage than KT-19. Therefore, we use the daily Level 4 Arctic sea and ice surface temperature reanalysis with a resolution of $0.05\times0.05°$ (Nielsen-Englyst et al., 2023), which matches the AMSR2 satellite footprint size, hereafter referred to as NE23. The product derives daily gap-free sea and ice surface temperatures from clear-sky thermal infrared satellite observations sensitive to the upper few millimeters of the snow or ice (Warren, 1982) and passive microwave-based sea ice concentration.

> We use the nearest NE23 ice surface temperature pixel to the satellite footprint as input to the sea ice emissivity calculation for satellites.

**1.8. Line 169**

**RC:** *Is there any potential here for confirmation bias? I.e. if the multiyear ice concentration maps are derived based on satellite observations, and assumed emissivity spectra?*

**AR:** Yes, there might be some potential for confirmation bias when using multiyear ice concentration from passive and active microwave observations to explain signatures from airborne passive microwave observations. The product from the University of Bremen incorporates additional information to minimize the risk of false interpretation of similar microwave signatures of both major ice types. These are temperature-dependent corrections and ice drift corrections (Melsheimer and Spreen, 2023). Therefore, this product, in combination with visual airborne imagery, provides a robust approach to classify the observed ice floes in the marginal sea ice zone. Furthermore, different microwave frequencies are used to create the product compared to our airborne observations, i.e., 5.3 GHz for ASCAT (VV polarization, normalized to 40°) and AMSR2 (19 and 37 GHz, V and H polarization, 55°; Melsheimer and Spreen, 2022). We included this explanation in the revised manuscript.

> Although the multiyear ice concentration product incorporates microwave observations of AMSR2 and ASCAT that might correspond to those at MiRAC frequencies, the implemented temperature and drift corrections increase independence between multiyear ice concentration and MiRAC TB.

**1.9. Line 177**

**RC:** *What variability is typically observed within the ±2h window?*

**AR:** The temporal variability is hard to quantify for the swath data, where footprints are always at slightly different locations. Therefore, we opt for a qualitative discussion of this by showing the time series of the 89 GHz TB from MHS collocated with MiRAC's 89 GHz channel as an example (Fig. R1). The same analysis

was performed for the other satellite instruments, and similar results were obtained. We generally find low temporal variability within a 4-hourly time window, especially for AFLUX, where flights cover a small domain. Higher variability occurs during ACLOUD RF23 for MHS. This flight covers a large area, which causes an artificial temporal drift. We added the outcome of this analysis to the revised manuscript. The paragraph also includes an analysis of the sea ice drift based on RC2 on line 178.

> We ensure simultaneous observations by filtering collocations within a $\pm 2$ h window, which maximizes the number of satellite overpasses and minimizes the effects of sea ice drift. The sea ice drifts less than 2.5 km within 2 h in the study area during flight days based on National Snow and Ice Data Center (NSIDC) sea ice drift data (Tschudi et al., 2020). Moreover, the relatively long time window is not problematic as the analysis of satellite data reveals much higher spatial than temporal variability (not shown).

[Figure]

Figure R1: Temporal variability of the TB from MHS and MiRAC resampled to the MHS footprints at 89 GHz for all flights where MiRAC provided 89 GHz TBs. MiRAC and satellites are separated by 10 min for better visibility.

**1.10. Line 205**

**RC:** *It should be clarified whether this transmissivity refers to the layer between the aircraft and the surface. And what about atmospheric emission?*

AR: We included the information that transmissivity refers to the layer between the aircraft and the surface. The atmospheric emission is contained in the final term of Eq. (3), i.e., the upwelling atmospheric radiation at the observation height $T_b^\uparrow$.

> The TB observed at aircraft or satellite height, denoted as $T_b$, is given by
>
> $$T_b = T_s \cdot e \cdot t + T_b^{\downarrow} \cdot t \cdot (1 - e) + T_b^{\uparrow}, \tag{1}$$
>
> with the surface emissivity $e$, surface temperature $T_s$, atmospheric transmissivity in viewing direction between the surface and the aircraft or satellite height $t$, downwelling atmospheric radiation at the surface $T_b^{\downarrow}$, and upwelling atmospheric radiation at the observation height $T_b^{\uparrow}$.

**1.11. Line 221**

**RC:** *It would still be interesting to compare the results for all 183GHz channels - albeit with increased uncertainties.*

**AR:** We agree that comparing emissivities at all 183 GHz channels is interesting. Therefore, we showed all 183 GHz channels that satisfy the surface sensitivity criterion in Fig. 3 of the manuscript. The figure shows that these channels provide similar emissivities, with slightly lower emissivities at the 183±2.5 GHz channel compared to 183±7.5 GHz channel. However, this difference lies within the uncertainty range and might be caused by the higher water vapor absorption at the inner channel and measurement uncertainties of atmospheric water vapor and temperature. These uncertainties directly propagate into the emissivity. The lack of direct measurements of the downwelling atmospheric radiation and related uncertainties in the atmospheric profile causes higher uncertainties at 183 GHz than during previous studies (e.g., observations with the airborne MARSS radiometer in Hewison et al., 1999, or Wang et al., 2017b). We provide all emissivities and estimated uncertainties in the published emissivity data to allow for different surface sensitivity thresholds for other applications. We justified our decision in the revised manuscript.

> Only 183 and 340 GHz observations during ACLOUD lie below the surface sensitivity threshold and are  excluded to avoid highly uncertain emissivity estimates.

**1.12. Line 225**

**RC:** *This is an underestimate in the case of strong surface inversions, which are common over sea ice.*

**AR:** See our response to the comment on line 153.

**1.13. Line 232**

**RC:** *How could the uncertainties be estimated for satellite observations?*

**AR:** The same approach can be applied to satellite observations. However, we do not perform this uncertainty estimation as reasoned in the revised manuscript.

> The uncertainty estimation is performed only on aircraft and not on satellite observations because the MiRAC channels already include most satellite channels. A notably higher emissivity uncertainty occurs for satellites near 183 GHz compared to MiRAC due to the higher atmospheric contribution.

**1.14. Line 246**

**RC:** *It is interesting to note that the results from the specular assumption appear more Gaussian. Why could that be?*

AR: We visualized the data of the histogram as a function of observed TB and research flight to better answer this question (Fig. R2a and b). The histograms shown in the manuscript are drawn below (Fig. R2c and d). The histogram shapes are partly determined by differences between the research flights and the dependence of the bias on the observed TB (for specular only). The difference between the flights causes the bimodality of the histogram for Lambertian reflection. The reason for the differences between the flights is likely a bias in the atmospheric profile. While the bias for RF08 is almost perfectly centered around 0 K for Lambertian reflection, the other flights show a slight positive bias. The positive bias could be related to an overestimation of the relative humidity or a misrepresentation of its vertical profile.

[Figure]

Figure R2: Difference between observed ($T_{b,\mathrm{obs}}$) and simulated 183.31±5 GHz TB ($T_{b,\mathrm{sim}}$) using 183.31±7.5 GHz emissivities under (a, c) Lambertian and (b, d) specular surface reflection during AFLUX. (a, b) Scatter plot between the difference and the observed TB. (c, d) Histogram of the difference with a TB bin width of 0.5 K.

**1.15. Figure 3**

**RC:** *I found this figure confusing - it took several readings before I understood it. It could also be expanded to a full page width.*

**AR:** This figure highlights the observations for an example flight transect. We will expand it to a full page in the revised manuscript. Also, we made an adjustment based on the second reviewer's comment to improve the visibility in panel (j) and added the new version to the RC2 response document (see comment on Figure 3 therein).

**1.16. Table 3**

**RC:** *How exactly is the mean relative uncertainty calculated?*

**AR:** The mean relative uncertainty is calculated by dividing the uncertainty by the emissivity and averaging this value over all samples. We clarified the table caption.

> Values denote the sample count (Cnt.), median (Mdn.), interquartile range (IQR), and  relative uncertainty averaged over all samples (Unc.).

**1.17. Figure 5**

**RC:** *I would also be interested to see the emissivity plotted as spectra for each cluster.*

**AR:** We agree that a representation as spectra is interesting (see Fig. R3). However, the 89 GHz channel is measuring at a different polarization and angle, so we decided not to include this representation in the manuscript.

**1.18. Line 377**

**RC:** *What are the implications of this assumption? Noting the results of Wang et al., 2017b differ from Harlow (2007): and Haggerty and Curry (2001), who found an increase in emissivity with frequency for sea ice between 150 and 220 GHz.*

**AR:** In the context of our comparison between airborne and satellite observations, we aimed to use a satellite channel less affected by atmospheric water vapor emission and as close to the MiRAC 183 GHz channel as possible. The goal of the sentence is to indicate that spectral emissivity gradients might be present. We have written this more clearly in the revised manuscript.

> The near-nadir (0 to 30°) 157 GHz MHS and 165.5 GHz ATMS channels are comparable to MiRAC's nadir 183 GHz channel. We compare these satellite channels instead of the 190.31 and 183.31±7 GHz channels due to their higher surface sensitivity and lower uncertainty, although spectral emissivity gradients might occur (e.g., Hewison et al., 2002). Other channel or instrument combinations differ in incidence angle or polarization, making footprint-level comparisons less meaningful. .

**1.19. Figure 6b**

**RC:** *This is a very useful result. But would it be better to divide by reflectivity (1-emissivity)? This might normalise the distribution.*

[Figure]

Figure R3: Sea ice emissivity spectra as a function of frequency. Thin lines show all samples and thick lines the cluster mean.

AR: Figure 6b highlights the dependence of the emissivity variability on the footprint size. The interquartile range acts as measure for variability. Dividing this measure by the mean or median reflectivity is more difficult to interpret, i.e., lower values would occur for lower emissivity compared to a higher emissivity with the same underlying interquartile range. This makes it difficult to compare different channels. Therefore, we show the emissivity interquartile range without applying a normalization by reflectivity.

**1.20. Line 396**

**RC:** *How exactly are the observations averaged to ensure equal spatial sampling?*

AR: The temporal and spatial collocation criteria are described in Sect. 2.5. Temporal alignment is ensured through the $\pm 2$ h time window and spatial alignment by the sensor-dependent distance threshold and a certain number of MiRAC observations within that distance. We simply average all MiRAC footprints that are collocated with one satellite footprint. We do not perform a weighted average because MiRAC does not cover the entire footprint area, and thus, little or no improvement is expected from a weighted mean. We modified

the sentence by adding a reference to the collocation section and replacing "equal" with "comparable."

> The MiRAC observations are averaged to the collocated footprints of each satellite instrument to ensure  comparable spatial sampling (see Sect. 2.5).

**1.21. Figure 7**

**RC:** *How exactly is the MiRAC emissivity resampled to satellite footprints?*

**AR:** See our response to the comment on line 396.

**1.22. Figure 7**

**RC:** *It would also be useful to plot the IQR for the emissivity derived from MiRAC and satellites independently. This figure could also be expanded to full page width.*

**AR:** The interquartile range is only available for MiRAC. Here, it is used as a measure of hectometer-scale emissivity variability within a single satellite footprint. This is different from Fig. 6, where the IQR was calculated over all samples. We also performed the same calculation here and provided the results in Tab. 4 and 5 for MiRAC (after averaging to satellite footprint), MHS, and ATMS.

**1.23. Line 401**

**RC:** *It is not clear how the Lambertian assumption introduces a bias less than 2% for MiRAC.*

**AR:** The statement is based on the comparison of 89 GHz emissivities under specular and Lambertian reflection (Fig. R4). The figure shows the percentage of specular emissivities exceeding Lambertian emissivities for the different frequencies for ACLOUD and AFLUX. It shows that the difference increases with decreasing emissivity and increasing atmospheric opacity. Similar results are described by Matzler (2005) and Karbou and Prigent (2005). For the reader, we extracted the relevant information from this figure, i.e., the percentage by which a fully specular emissivity would exceed the fully Lambertian emissivity provided in all our figures. This is only done for 89 GHz because we do not have polarization information from space that hints at specular contributions to the reflection and found that Lambertian emissivity is more consistent. However, we made a mistake and extracted only the values for AFLUX, not for ACLOUD. We corrected this in the revised version. For ACLOUD, the uncertainty due to surface reflection could exceed the emissivity uncertainty that we estimated. This is due to the higher optical depth compared to AFLUX. At the same time, it is unlikely that sea ice is purely specular at 89 GHz (e.g., a specularity parameter of 0.5 was found over Antarctica in summer at 50 GHz in Guedj et al., 2010, which also shows good results over sea ice in Bormann, 2022) and the true uncertainty is lower. An exact quantification is impossible at this point. We modified line 255 in Sect. 3.3 and line 401 in Sect. 5.3.

Change of line 255:

> For MiRAC observations at 89 GHz , fully specular emissivities  exceed fully Lambertian emissivities by about 6 to 2 % during ACLOUD and 3 to 1 % during AFLUX for Lambertian emissivities from 0.6 to 0.8 . This emissivity uncertainty is in the order of or lower than the uncertainty due to the surface temperature assumption since sea ice is not fully specular at 89 GHz (Bormann, 2022) .

Change of line 401:

As expected, the 89 GHz emissivity shows a polarization signal of about 0.1.  This indicates a specular contribution to the surface reflection and an underestimation of the emissivity under purely Lambertian reflection at 89 GHz for MiRAC (see Sect. 3.3).

[Figure]

Figure R4: Relative difference between specular and Lambertian emissivity as a function of Lambertian emissivity for (a) ACLOUD and (b) AFLUX and the four channels.

**1.24. Table 4**

**RC:** *Is the relative bias here calculated from the mean difference? It does not seem to match the difference of the median values.*

AR: Yes, the relative bias is calculated from the mean difference normalized element-wise by the MiRAC emissivity. It mostly matches the tendency of the median value, but not always as the emissivity distribution is slightly skewed (see Fig. 8b for MHS and ATMS).

**1.25. Figure 8**

**RC:** *This is potentially a very useful figure, but is confusing to interpret - especially the labelling on the x-axis should be improved. It may also help to more clearly distinguish V & H polarisation.It would be better supplemented by a table of values.*

1. *Why are multiple values shown for 243GHz (2) 340GHz (3)?*

2. *Any idea why the 89V results for AMSR2 are out-of-family?*

> *3. AMSR2 and SSMIS are conical scanners, with V and H polarisations, not QV as shown.*
>
> *4. Why are no ATMS results are given for ACLOUD?*

AR: We updated Fig. 8 of the manuscript. We rotated the x-axis labels and removed the transparency of the count bars. We did not indicate the V/H polarization with, e.g., a hatch pattern because they are already clearly separated at 89 GHz, and it requires too many different patterns to indicate QV, QH, and nadir. Therefore, the polarizations are indicated only on the x-axis. We do not think that values in a table provide additional value to this figure, as variability is a key element that we want to highlight here. The following replies refer to the four specific questions.

1. Multiple boxes are shown for 243 and 340 GHz due to the collocation method that matches MiRAC observations to each satellite observation. One satellite observation consists of a specific footprint and channel. The separation by channel was performed because MiRAC's 89 GHz is not available for low flight altitudes (see Sect. 2.5), and we want to ensure comparable spatial sampling, at least for the same spectral bands, for better comparison. We consider the 243 and 340 GHz channels as part of the >100 GHz band and, therefore, show the same collocation as for the 183 GHz MiRAC channel. Therefore, we also do not show AMSR2 for MiRAC channels above 100 GHz. The distributions for the different sensors are similar, which is an encouraging result. Only for AMSR2, we find differences at 89 GHz due to the low count of AMSR2 footprints during ACLOUD (see question 2).

2. The AMSR2 89V results differ because the sensor only covers the western part of ACLOUD RF23 after applying the collocation method. Only 23 footprints collocate with MiRAC's 89 GHz channels, as indicated by the low count in the bar plot of Fig. 8a. The two overpasses are at 09:46 and 11:25 UTC, respectively. The first MiRAC sample is at 11:35 UTC. SSMIS covers also the eastern part, which has a lower emissivity at vertical polarization.

3. We calculated QV from H and V for AMSR2 and SSMIS at the given incidence angle for comparison with MHS and ATMS, although the incidence angle differs here with comparable footprint sizes. This is mentioned also in line 401.

4. No ATMS results can be shown for ACLOUD because no ATMS footprint passes the collocation criteria. This has been mentioned earlier in line 183. We added a note to the caption of Fig. 8.

> The 88 to 92 GHz satellite footprint count might be lower than above 150 GHz because satellite footprints are excluded if the nearest MiRAC channel contains no emissivity. Note that no ATMS overpass occurred during ACLOUD.

**1.26. Line 465**

RC: *How much difference is expected from nadir to 25°? I would not expect much, following Hewison and English (1999): Airborne Retrievals of Snow and Ice Surface Emissivity at Millimetre Wavelengths. IEEE Trans. Geosci.Remote Sensing, Vol.37, No.4, 1999, pp.1871-1879, doi:10.1109/36.774700*

AR: The QH emissivity at 89 GHz of Fig. 5 in Hewison and English (1999) shows angular variation for all sea ice types. Between nadir and 25°, this is about 0.02 for nilas and similar for other ice types. We expect similar differences for our airborne observations between nadir and horizontal polarization at 25° because the polarization difference observed by SSMIS and AMSR2 at 89 GHz is about 0.1 near 53° (Fig. 8). Therefore, we consider the different polarization and viewing angle of MiRAC-A and MiRAC-P observations as one of the limitations of this study.

**1.27. Line 263**

**RC:** *Typo: Word -> World*

AR: Done.

> Notably, surface structural variations from 3 to 4 km suggest the presence of young ice, which defines ice in the transition stage between nilas and first-year ice ( World Meteorological Organization, 2014), possibly formed within leads among the thicker multiyear ice.

**1.28. Line 473**

**RC:** *I suggest "resolve" instead of "capture", noting it will still affect the mean emissivity.*

AR: Done.

> Spatial resolution: MiRAC's hectometer scale may not  resolve smaller sea ice features such as ridges or melt ponds, which could influence emissivity.

---

## Author Comment (AC2)

**Authors' Response to Reviews of**

**Assessing the sea ice microwave emissivity up to submillimeter waves from airborne and satellite observations**

Nils Risse, Mario Mech, Catherine Prigent, Gunnar Spreen, and Susanne Crewell

*The Cryosphere,* https://doi.org/10.5194/egusphere-2024-179
* * *
RC: *Reviewers' Comment*,    AR: Authors' Response,    □ Manuscript Text

**2.   RC2, Dr. Melody Sandells**

**2.1.   General comment**

RC: *This manuscript addresses the uncertainty in sea ice microwave emissivity representation for numerical weather prediction applications. Quantification of the sea ice contribution to satellite signals is crucial in order to separate surface and atmospheric contributions to satellite signals. This paper identifies sea ice type from microwave emissivity spectra via K-means clustering, demonstrates appropriateness of Lambertian scattering assumptions and investigates scaling issues by resampling airborne observations to satellite resolution and comparing with satellite data, considering resolution, incidence angle, polarisation as well as frequency. This manuscript is well-written and robust with justified assumptions and demonstrates that representative emissivity based on sea ice type is a reasonable approach and consequently that the spatial variability in sea ice properties must be accounted for. This manuscript is suitable for publication with minor amendments, and the following points considered in discussion:*

AR: The authors would like to thank Dr. Melody Sandells for their valuable time reviewing this manuscript and providing constructive feedback. We have carefully considered all comments and provided author responses below.

**2.2.   Line 39-41**

RC: *Please expand on the Hewison study to discuss what was found and how it relates to these results. This is already included around line 280, but what is needed here is to highlight the new frequencies in this approach, particularly given that the higher frequencies are more sensitive to surface type.*

AR: We extended the description of Hewison et al. (2002) by adding two sentences on their results, i.e., new ice, first-year ice, and multiyear ice emissivity spectra, with a focus on the higher frequencies.

> Hewison et al. (2002) calculated nadir emissivities  from 24 to 183 GHz of sea ice with different development stages from new to multiyear ice with similar instrumentation as in Hewison and English (1999). New ice emissivities were highest and slightly decreased from 0.95 at 89 GHz to 0.9 at 183 GHz. First-year ice emissivities decreased from 24 to 157 GHz and slightly increased from 157 to 183 GHz. This emissivity increase towards higher frequencies was also found for multiyear ice . Haggerty and Curry (2001) observed first-time emissivities up to 243 GHz at nadir at about 1 km$^2$ resolution.

**2.3. Line 79-80**

**RC:** *Just to link with the previous section state that the Polar 5 carried the MiRAC and KT-19 instruments (see comment for line 156).*

**AR:** We added another sentence to mention the remote sensing instrumentation on board Polar 5.

> The research flights (RFs) with the Polar 5 aircraft (Wesche et al., 2016) from the Alfred Wegener Institute Helmholtz Centre for Polar and Marine Research (AWI) covered the Fram Strait northwest of Svalbard, Norway (Fig. 1). *Polar 5 carried the microwave package MiRAC, the thermal infrared sensor KT-19, and a visual camera, among other instruments.*

**2.4. Line 81**

**RC:** *'1). Various sea ice characteristics were observed... ': specify this is from the airborne observations as no in situ measurements were made.*

**AR:** We modified the sentence to avoid confusion with in situ measurements.

> Various sea ice characteristics were observed *with Polar 5* during ACLOUD, i.e., RF23 on 25 June and RF25 on 26 June 2017, and AFLUX, i.e., RF08 on 31 March, RF14 on 8 April, and RF15 on 11 April 2019, under clear-sky conditions over sea ice suitable for emissivity estimation.

**2.5. Line 84**

**RC:** *Is ACLOUD firstyear, multiyear or a mix or ice types? The description for AFLUX was very helpful – please include a comparable description for ACLOUD.*

**AR:** The sea ice type retrievals, which are based on microwave observations, provide no information during the melt season, because the backscatter and emission signals of first- and multiyear ice become more similar (e.g., Lindell and Long, 2016). For the Arctic, the multiyear ice concentration products are typically available from May to October. The AMSR2/ASCAT product used here provides a classification until 8 May 2017, which is 48 days before the first ACLOUD flight where we derived emissivities. Therefore, no sea ice type was mentioned for the two ACLOUD flights. Instead, we mention the presence of melt ponds and open water in between individual ice floes. We clarify this by adding "wintertime" to the multiyear ice product description in Sect. 2.4 (line 169).

Lindell DB, Long DG. Multiyear Arctic Ice Classification Using ASCAT and SSMIS. Remote Sensing. 2016; 8(4):294. https://doi.org/10.3390/rs8040294

> Finally, three data products add surface information, i.e., daily sea ice concentration maps of the University of Bremen with $6.25 \times 6.25$ km$^2$ resolution based on AMSR2 (Spreen et al., 2008), daily *wintertime* multiyear ice concentration maps of the University of Bremen with $12.5 \times 12.5$ km$^2$ resolution based on AMSR2 and the Advanced Scatterometer (ASCAT; Melsheimer and Spreen, 2022), and Sentinel-2B Level 2A visual images with $20 \times 20$ m$^2$ resolution (European Space Agency, 2021).

**2.6. Line 88**

**RC:** *How was the integrated water vapour measured? Add a link to (presumably) section 2.4.*

**AR:** The integrated water vapor was derived from the in situ atmospheric profiles from dropsondes, radiosondes,

and the aircraft's nose boom as described in Sect. 2.4. These profiles are also used for the emissivity calculation. We added a link to Sect. 2.4 in the revised manuscript.

> The integrated water vapor, derived from in situ observations (see Sect. 2.4), is about 10 to 10.3 $\mathrm{kg\,m^{-2}}$ during the two ACLOUD flights and 1.3 to 2 $\mathrm{kg\,m^{-2}}$ during the three AFLUX flights, which indicates reduced water vapor emissions and high atmospheric transmissivity during AFLUX.

**2.7. Figure 1**

**RC:** *Please use a different colour scale to distinguish between RF23 and RF25 and between RF14 and RF15. Perhaps use different line thicknesses or line type.*

**AR:** We changed the line colors and widths to improve the visual clarity.

**2.8. Table 1**

**RC:** *Add 'Passive' into the table caption and consider including the KT19 sensor characteristics.*

**AR:** We added passive into the table caption.

> Specifications of the passive MiRAC-A channel and MiRAC-P channels.

**AR:** We excluded KT-19 from the table to solely list passive microwave channels. However, we agree that it is useful to compare the incidence angle and field of view information of these sensors. The information on the incidence angle is currently not mentioned and we added it in Sect. 2.4, line 156 (see the response to the comment on line 156).

**2.9. Line 145**

**RC:** *It would be useful to remind the reader here that MiRAC 89GHz is only available at 25 deg.*

**AR:** We clarified this in the text.

> However, MiRAC's 89 GHz channel with, which measures under horizontal polarization at 25°, is not directly comparable with the satellite channels because MHS and ATMS measure mostly vertically polarized TB near this incidence angle, and SSMIS and AMSR2 measure at higher incidence angles.

**2.10. Line 156**

**RC:** *This is the first mention of the KT-19 sensor (apart from line 119) – presumably also on the Polar 5, but please clarify.*

**AR:** Yes, the KT-19 is also on board Polar 5 and we added it to the sentence. This revised sentence also includes parts of the comment on Table 1 to avoid duplicate versions.

> The airborne KT-19 on board *Polar* 5 provides infrared TBs integrated over the atmospheric window from 9.6 to 11.5 µm with 1 s resolution under an opening angle of 2° at nadir.

**2.11. Line 178**

**RC:** *What is the estimated drift rate and how was this determined?*

AR: Climatological studies such as Kaur et al. (2018) found sea ice drift rates of about 8 to 10 km/d for the Fram Strait region. We also looked at daily sea ice drift data from the National Snow and Ice Data Center(NSIDC; Tschudi et al., 2020). We added this to the revised manuscript with an analysis of the temporal variability based on the RC1 comment on line 177.

Kaur S, Lukovich JV, Ehn JK, Barber DG. Higher-order statistical moments to analyse Arctic sea-ice drift patterns. Annals of Glaciology. 2020;61(83):464-471. doi:10.1017/aog.2021.6

Tschudi, M. A., Meier, W. N., and Stewart, J. S.: An enhancement to sea ice motion and age products at the National Snow and Ice Data Center (NSIDC), The Cryosphere, 14, 1519–1536, https://doi.org/10.5194/tc-14-1519-2020, 2020.

> We ensure simultaneous observations by filtering collocations within a $\pm 2$ h window, which maximizes the number of satellite overpasses and minimizes the effects of sea ice drift. The sea ice drift during the flights is less than 1 kmh$^{-1}$ based on data from the National Snow and Ice Data Center (NSIDC; Tschudi et al., 2020), and spatial variability exceeds temporal variability (not shown).

**2.12. Line 184**

RC: *Consider moving 'during ACLOUD (AFLUX)' to after 'overflights' so the meaning is better conveyed before the brackets are used. Could the information in this section be better displayed as a table?*

AR: We rearranged the sentence. Yes, the information on the number of satellite overflights and collocated footprints is more useful inside a table. Also, the number of satellite footprints without channel failure, e.g., 150 GHz of DMSP-F18/SSMIS (mentioned in Sect. 2.3), is important. We already provided this information in the results section combined with the emissivity statistics (see Tab. 4 and 5 for MHS and ATMS during AFLUX and Fig. 8 for all channels).

AR: Also, we noticed a mistake in the code where the distance threshold to the shoreline was 7.5 km instead of 8 km for MHS, ATMS, and SSMIS. This lead to the exclusion of one SSMIS footprint. We modified the number in the following sentence in the revised manuscript. The Fig. 8 of the manuscript will also be updated, but the change is hardly visible.

> The number of satellite overflights during ACLOUD (AFLUX) with collocated footprints from MHS, ATMS, SSMIS, and AMSR2 is 15 (23), 0 (8), 11 (26), and 2 (9) , respectively. We matched channels near 89 GHz with MiRAC-A and above 100 GHz with MiRAC-P. The number of satellite footprints collocated with MiRAC at 89 GHz during ACLOUD (AFLUX) is 87 (86), 0 (34),  107 (175), and 23 (159) for MHS, ATMS, SSMIS, and AMSR2, respectively.

**2.13. Line 255**

RC: *Are the numbers in brackets for ACLOUD or AFLUX? In general it's better to write this out in full for ease of reading.*

AR: This sentence describes the difference between specular and Lambertian emissivities as a function of Lambertian emissivity. We modified this sentence along with line 401 based on the comment of RC1 on line 401 and provided both revisions under that comment.

**2.14. Line 262**

**RC:** *'We observe predominantly snow-covered sea ice over the transect's initial 7 km' – is this from right to left as per Westerly flight, or left to right as per numbering in Fig 3?*

**AR:** This sentence refers to the part from 0 to 7 km, as drawn in Fig. 3. To avoid confusion with the flight direction, we clarified the sentence.

> We observe predominantly snow-covered sea ice  from 0 to 7 km.

**2.15. Line 263**

**RC:** *Typo: 'Word' -> 'World'*

**AR:** Done.

**2.16. Line 290**

**RC:** *'The ±8 K surface temperature uncertainty causes the highest emissivity uncertainty for all channels.' Where is this demonstrated?*

**AR:** This is provided as additional information and is not shown in Fig. 3, which indicates only the total error. We modified the sentence now by adding "not shown". Generally, this result originates from the error calculation that we describe in Sect. 3.2.

> The $\pm 8$ K surface temperature uncertainty causes the highest emissivity uncertainty for all channels (not shown).

**2.17. Line 304**

**RC:** *'and we found no significant changes in the shapes of the histograms (not shown)'. What statistical test was used?*

**AR:** This statement is based on a comparison of the emissivity distributions with and without matching the footprints of MiRAC-A and -P. We performed a Kolmogorov–Smirnov test, which indicated that the samples do not originate from the same distribution. However, one must consider that emissivity biases vary regionally, i.e., temperature gradients in the snow and sea ice, air temperature biases, and relative humidity biases. This causes differences in the emissivity distributions for our limited number of flights. We modified the sentence of the revised manuscript and removed the word "significant."

> The 89 GHz and 183 to 340 GHz histograms include different samples due to the exclusion of low flight altitudes at 89 GHz, which introduces a potential inconsistency (Table 3). Therefore, we compared these histograms with those from instantaneous measurements where all channels sample the same sea ice, and we found no  changes in the shapes of the histograms that exceed the estimated emissivity uncertainties (not shown).

**2.18. Figure 3**

**RC:** *Please put this through a colour blind checker, particularly fig 3j, where it's hard to distinguish between 183 +/- 2.5 and 3.5 GHz bands.*

AR: We fixed this in the revised version.

**2.19. Figure 4**

RC: *Please use a different colour scheme to distinguish between the two ACLOUD flights.*

AR: We updated the plot with the new colors, as in Fig. 1.

**2.20. Line 351**

RC: *What test of significance was performed?*

AR: The word significant implies statistical tests and has been used in the wrong context here. We modified the sentence while retaining its meaning.

> However, the emissivity variability at both frequencies is still  notable and depends on the sea ice type, with the highest contrast between multiyear ice and nilas.

**2.21. Line 368**

RC: *'Hence, the satellite footprint contains mean conditions where significant small-scale variability averages out.' I am unsure what is meant by this and how it relates to the previous sentences – please could you clarify?*

AR: The sentence aimed at summarizing the findings from Fig. 6b of the manuscript, which shows that a high emissivity variability occurs on hectometer scales. This high emissivity variability reduces with increasing footprint sizes up to the satellite scale. We agree that the sentence is unclear and adjusted it to the following:

> Hence, the larger satellite footprint  averages out small-scale emissivity variations.

**2.22. Line 382**

RC: *'The limited spatial coverage of MiRAC causes slightly higher emissivity variability compared to MHS and ATMS, as MiRAC only captures a narrow strip of the satellite footprint'. Why this rather than simply the higher resolution of MiRAC?*

AR: The sentence was not precise and we modified it to convey the message that some areas are not well represented due to the incomplete coverage of the satellite footprint by MiRAC. The higher resolution of MiRAC will not be effective anymore after averaging it onto the satellite footprint.

> The limited spatial coverage of MiRAC causes  deviations from MHS and ATMS, as MiRAC only captures a narrow strip of the satellite footprint, e.g., during AFLUX RF08 near 80.4° N, 5° E (Fig. 7a) leading to the highest emissivity bias (Fig. 7d).

**2.23. Figure 6**

RC: *Does the cluster colour scheme relate to the emissivity colour palette?*

AR: The cluster colors are extracted from the same color map that is used for the emissivities. The cluster numbers were sorted such that the emissivity increases from cluster 1 to 4 at 89 GHz (see Fig. 5). Therefore, the dark

color of cluster 1 corresponds to lower emissivities and the bright color of cluster 4 corresponds to higher emissivities.

**2.24. Line 396**

RC: *As the satellites have different footprints 'equivalent spatial sampling' may be better than 'equal spatial sampling'*

AR: We modified the respective sentence.

> The MiRAC observations are averaged to the footprints of each satellite instrument to ensure  equivalent spatial sampling.

**2.25. Figure 7(m)**

RC: *What is in the wider satellite footprint that is causing the higher emissivity in the western tip?*

AR: We identified a mistake in Fig. 7. The third column does not show channel 2 from MHS (157 GHz) and channel 17 from ATMS (165.5 GHz) as also indicated in the label, but channel 5 from MHS (190.31 GHz) and channel 18 from ATMS (183.31±7.5 GHz). The feature is less pronounced in the 157 and 165.5 GHz channels of MHS and ATMS and likely relates to water vapor or temperature gradients that are not represented by our in situ profile. Another reason for the emissivity difference between MiRAC and MHS/ATMS are lower NE23 surface temperatures compared to KT-19

**2.26. Line 416**

RC: *'Additionally, AMSR2 shows higher variability due to its smaller footprint than SSMIS'. This conflicts with ACLOUD IQR being smaller at Vpol for AMSR2 than SSMIS in Fig 8a.*

AR: We explain this discrepancy by the few collocated footprints of AMSR2 with MiRAC during ACLOUD RF23 (see the low count in Fig. 8a). For AFLUX, the number of footprints for SSMIS and AMSR2 is similar. We modified the sentence to indicate this better.

>  For AFLUX, where the footprint count of SSMIS and AMSR2 is comparable, AMSR2 shows higher variability due to its smaller footprint than SSMIS.

**2.27. Line 469**

RC: *'Surface temperature assumption: Using the surface skin temperature instead of the emitting layer temperature imposes a frequency-dependent bias on the emissivity during AFLUX'. How much does this assumption influence the conclusion that the emissivity spectra are relatively flat?*

AR: We expect differences in the emitting layer temperature, especially between 89 and 150 GHz based on calculated penetration depths in Tonboe et al. (2006) and simulated emitting layer temperatures in Tonboe (2010). As penetration depth decreases toward higher frequencies, the emitting layer temperature lies closer to the skin temperature. Therefore, the frequency-dependent temperature bias would decrease towards 340 GHz. However, the effect of surface temperature on the emissivity is frequency-dependent as well for the method we use, with higher effects at higher frequencies. We expect that the bias lies within the uncertainties we provide in Tab. 3. Therefore, it would not largely affect the assumption that the emissivity spectra are relatively flat.